# Responses of sub-ice platelet layer thickening rate and frazil ice concentration to variations in Ice Shelf Water supercooling in McMurdo Sound, Antarctica

Chen Cheng[1,2,3], Adrian Jenkins[2], Paul R. Holland[2], Zhaomin Wang[4], Chengyan Liu[4,1,3], and Ruibin Xia[1,3]

[1]Polar Climate System and Global Change Laboratory, Nanjing University of Information Science & Technology, Nanjing, 210044, China
[2]British Antarctic Survey, Cambridge, CB3 0ET, UK
[3]School of Marine Sciences, Nanjing University of Information Science & Technology, Nanjing, 210044, China
[4]College of Oceanography, Hohai University, Nanjing, 210098, China

*Correspondence to*: Zhaomin Wang (zhaomin.wang@hhu.edu.cn)

**Abstract.** Persistent outflow of supercooled Ice Shelf Water (ISW) from beneath McMurdo Ice Shelf creates a rapidly growing sub-ice platelet layer (SIPL) having a unique crystallographic structure under the sea ice in McMurdo Sound, Antarctica. A vertically-modified frazil-ice-laden ISW plume model that encapsulates the combined nonlinear effects of the vertical distributions of supercooling and frazil concentration on frazil ice growth is applied to McMurdo Sound, and is shown to reproduce the observed ISW supercooling and SIPL distributions. Using this model, the dependence of SIPL thickening rate and depth-averaged frazil ice concentration on ISW supercooling in McMurdo Sound is investigated, and found to be predominantly controlled by the vertical distribution of frazil concentration. The complex dependence on frazil concentration highlights the need to improve frazil ice observations within the sea ice-ocean boundary layer in McMurdo Sound.

## 1 Introduction

Ice shelf basal melting removes more mass from the Antarctic Ice Sheet than iceberg calving does, but the three largest ice shelves, Filchner-Ronne, Ross, and Amery, contribute only 18% of the net meltwater flux (Rignot et al., 2013). That is because the seawater-filled cavities beneath those ice shelves are dominated by High Salinity Shelf Water that has a potential temperature at or near the surface freezing point. Ice shelf basal melting occurs at depth, because the freezing point temperature is lower under elevated pressure, and results in the formation of Ice Shelf Water (ISW), characterized by potential temperatures below the surface freezing point. When the buoyant ISW ascends along the ice shelf base, the pressure relief causes it to become supercooled in situ, a necessary condition for ice crystals to persist in suspension. Those disk-shaped frazil ice crystals accumulate under the ice shelves, leading to the formation of marine ice that is thicker and more localized than would be possible through direct freezing at the ice shelf base (Morgan, 1972; Oerter et al. 1992; Fricker

et al., 2001; Holland et al., 2007, 2009). Occasionally, frazil ice crystals bathed in supercooled ISW are also carried out beyond the ice shelf front and precipitated under adjacent sea ice, forming an unconsolidated, porous, sub-ice platelet layer (SIPL) (Gow et al., 1998; Hunkeler et al., 2016; Langhorne et al., 2015; Leonard et al., 2006; Robinson et al., 2014). SIPL not only harbours some of the highest concentrations of sea ice algae on Earth (Arrigo et al., 2010) but also contributes to the

sea ice thickness when the water within the pores of SIPL freezes, due to heat loss to the atmosphere, to become incorporated platelet ice (Smith et al, 2001). Therefore, SIPL should not be ignored when investigating sea ice thickness near an ice shelf front.

Owing to the paucity of direct observation, our understanding of the evolution of frazil-ice-laden ISW relies heavily on

numerical models. Those models are mostly derived from plume theory (Holland and Feltham, 2006; Jenkins and Bombosch, 1995; Rees Jones and Wells, 2018; Smedsrud and Jenkins, 2004), but include three-dimensional ocean circulation models (Galton-Fenzi et al., 2012), and have been widely applied to assess the marine ice beneath Filchner-Ronne (Bombosch and Jenkins, 1995; Holland et al., 2007; Smedsrud and Jenkins, 2004), Larsen (Holland et al., 2009) and Amery ice shelves (Galton-Fenzi et al., 2012), and SIPL under the sea ice in McMurdo Sound (Hughes et al., 2014, hereinafter HU14). To date,

all the ISW plume models mentioned above have been depth-integrated, and all the scalar quantities, i.e., potential temperature, salinity, and frazil concentration in those models are treated as vertically-uniform. The well-mixed potential temperature and salinity have been validated by borehole observations beneath the Amery Ice Shelf (Herraiz-Borreguero et al., 2013) and under the sea ice in McMurdo Sound (Robinson et al., 2014; HU14). Although there are no observations of the vertical profile of frazil ice concentration, it is unlikely to be vertically uniform because the buoyant rise of the crystals will

counteract the turbulent diffusion that tends to homogenise the other properties. Recently, Cheng et al. (2017) showed that adopting an approach in which the frazil ice growth is calculated using a vertically-uniform frazil concentration results in substantial underestimation of marine ice production underneath the western side of Ronne Ice Shelf. Idealized one dimensional models confirm that the vertical distribution of frazil concentration cannot remain well-mixed in the upper layers of the ocean (Svensson and Omstedt, 1998) and beneath ice shelves (Holland and Feltham, 2005). Consequently,

earlier assessments of either marine ice or SIPL production in the aforementioned areas may need to be re-evaluated.

McMurdo Sound, located in the southwestern Ross Sea (Fig. 1), is characterized by significant ISW outflow, arguably one of the most comprehensively observed ISW plumes available (HU14; Langhorne et al., 2015; Robinson et al. 2014). A prominent SIPL forms in the central-western sound (Dempsey et al., 2010); the maximum (area-averaged) observational

first-year sea ice and SIPL are 2.5 (2) and 8 (3) m as determined from drill-hole measurements adjacent to McMurdo Ice Shelf front between late November and early December in 2011 (Fig. 9 in HU14). The thin (~20 m) McMurdo Ice Shelf front allows the ISW outflow to be delivered to the ocean surface without mixing with warmer ambient waters (Robinson et al., 2014). The study documented in HU14 was the first to apply the steady, one-dimensional frazil-ice-laden ISW plume model developed by Smedsrud and Jenkins (2004) to McMurdo Sound, although a constant ISW plume thickness was used.

McMurdo Sound therefore seems an ideal setting in which to apply and evaluate the new vertically-modified ISW plume model proposed by Cheng et al. (2017), which includes time dependence and two horizontal dimensions. The main objective is to explore possibility of finding the quantitative relationship between SIPL thickening rate and ISW supercooling. Establishing such a relationship is of significance to the assessment of total sea ice thickness, and thus the oceanic heat flux associated with SIPL, in McMurdo Sound and elsewhere.

Here we first analyze the combined nonlinear effects of the vertical distributions of supercooling and frazil concentration on the suspended frazil ice growth rate in a supercooled ISW plume, and compare results with those obtained with a commonly-used, depth-averaged formulation. Then, we evaluate the performance of the vertically-modified ISW plume model in reproducing the observed ISW supercooling and SIPL distribution to show the importance of considering the combined nonlinear effects. Finally, we conduct 211 sensitivity simulations with the purpose of quantitatively establishing the response of SIPL thickening rate as well as the frazil ice concentration to variations in ISW supercooling in McMurdo Sound.

## 2 Physically-based formulation for frazil ice growth rate

The growth rate of suspended frazil ice controls both the dynamic and thermodynamic evolution of ISW plumes and the accretion of ice crystals beneath ice shelves (Cheng et al., 2017; Holland and Feltham, 2006; Smedsrud and Jenkins, 2004) and sea ice (HU14). The frazil ice growth rate is found to be proportional to the following integral expression once a number of physical parameters within the commonly-used formulation of Jenkins and Bombosch (1995) are merged:

$$I_{gr} = \int_0^1 T_{SC}\, c_i(\sigma)d\sigma, \quad T_{SC} = T_f(\sigma,S) - T, \tag{1}$$

where $\sigma \in [0, 1]$ is the relative vertical coordinate, with 0 and 1 respectively corresponding to the upper ice-plume and lower plume-ambient water interfaces, $T$ and $S$ are respectively the plume's potential temperature and salinity, vertically well-mixed within the plume, $c_i$ is the vertically-distributed, in this study, volumetric frazil concentration within the plume, $T_{SC}$ and $T_f$ are respectively the supercooling level (positive for supercooling) and local freezing point. Because of the well-known linear decrease in $T_f$ with increasing water depth, $T_{SC}$ also varies linearly with depth, transitioning from supercooling to overheating as $\sigma$ increases (Figs. 2a and 3). The corresponding transition height at which $T_{SC} = 0$ is defined by supercooled thickness $D_{SC} = \sigma_{SC}D$ where $\sigma_{SC}$ and $D$ are respectively supercooled fraction and total ISW plume thickness.

In earlier ISW plume models, because $c_i$ is treated as vertically-uniform, the integral of (1) can be represented by the product of the depth-averaged values $T_{SC}^{0.5}$ (0.5 means at mid-depth) and $C_i$. Thus, we refer to these ISW plume models as vertically-uniform (VU). It is worth mentioning that in order to take the supercooling into account when $\sigma_{SC}<0.5$, HU14 integrated $T_{SC}$ over the supercooled part only without introducing any frazil ice melting. However, in this study, we will demonstrate that the important role of frazil ice melting in the lower, overheated part of the plume cannot be ignored.

The vertical distribution of frazil concentration, in reality, much like the concentration of suspended sediment (Cheng et al., 2013, 2016), should be vertically non-uniform, with higher concentrations near the ice shelf/sea ice base. Considering only the balance between the buoyant-rise-induced vertical advection and turbulent diffusion terms, the governing equation for frazil concentration can be written as

$$\frac{d}{d\sigma}\frac{K}{D}\frac{dc_i}{d\sigma} + w_i\frac{dc_i}{d\sigma} = 0,$$

where $w_i$ is the frazil ice rise velocity, determined by ice crystal size, $K$ is the vertical frazil concentration diffusion coefficient, which can be parameterized as vertically constant (Cheng et al., 2013, 2016):

$$K = \frac{1}{6}\kappa u_* D,$$

where $\kappa = 0.4$ is von Karman's constant, $u_* = \sqrt{C_d}U$ is the friction velocity, related to the turbulent intensity within the ISW plume, $C_d$ is the basal drag coefficient, $U = \sqrt{(U_p + U_a)^2 + (V_p + V_a)^2 + U_t^2}$ is the total flow speed, $U_p(U_a)$ and $V_p(V_a)$ are the depth-averaged ISW plume (ambient current) speed in the $x$ and $y$ directions respectively, $U_t$ is the root-mean square tidal speed. Using a zero net flux condition in the equilibrium state at the lower boundary of the plume, i.e.,

$$\frac{K}{D}\frac{dc_i}{d\sigma} + w_i c_i = 0, \text{ for } \sigma=1$$

and a Dirichlet boundary condition at the upper boundary, i.e.,

$$c_i = c_{i,b}, \text{ for } \sigma=0$$

where $c_{i,b}$ is the frazil concentration at the ice shelf/sea ice base, the vertical exponential profile for the equilibrium frazil concentration can be readily obtained (Cheng et al., 2017):

$$\frac{c_i(\sigma)}{c_{i,b}} = exp(-6Z_*),$$

where $Z_* = w_i/\kappa u_*$ is the suspension index, otherwise known as the Rouse number. Integrating this exponential profile from $\sigma=0$ to 1, we finally obtain the relation between $c_i(\sigma)$ and $C_i$ as

$$\frac{c_i(\sigma)}{C_i} = \frac{6Z_* exp(-6Z_* \sigma)}{1 - exp(-6Z_*)}. \tag{2}$$

As shown in Fig. 2a, the vertical distribution of frazil concentration is strongly controlled by $Z_*$. The gradient of the vertical distribution becomes greater with increasing $Z_*$, and a vertically-uniform frazil concentration distribution can only be achieved as $Z_*$ approaches 0. While low values of $Z_*$ are attainable with strong currents, those conditions also reduce the tendency for frazil to precipitate and contribute to SIPL formation [see Eq. (3) below]. Therefore, we expect a non-uniform vertical distribution of frazil wherever there is active formation of SIPL. Accordingly, Cheng et al. (2017) introduced (2) into (1), and as a result significantly improved the simulated pattern of marine ice growth under the western side of Ronne Ice Shelf, compared with the VU and satellite-derived (Joughin and Padman, 2003) results. Hereinafter, we refer to this

vertically-modified ISW plume model as VM. To conclude, the only difference between VM and VU models is whether the vertical distribution of frazil ice concentration is introduced.

The dependence of the integral value of $I_{gr}$ on $Z_*$ under specified conditions of supercooling (Fig. 2a) is shown in Fig. 2b, where $D_{SC} = 50$ m (a value within the calculated range for the standard run, Fig. 1) in all the cases. It can be seen that the integral value increases nonlinearly with $Z_*$. The critical $Z_*$ that represents the transition from frazil ice melting ($I_{gr} < 0$) to freezing ($I_{gr} > 0$) decreases as the supercooled part of ISW plume increases. In contrast, owing to the neglect of vertical variation in $c_i$, the integral values calculated using the VU formulation are constant, leading to transitions from overestimation of frazil ice growth to underestimation, compared with VM, as $Z_*$ increases. Only if the ISW plume is fully supercooled ($\sigma_{SC} =1$) and $Z_*$ is close to 0 are the integral values of $I_{gr}$ calculated by VU and VM formulations equal (star in Fig. 2b). These features are illustrated in Fig. 2a: for given supercooling, if $Z_*$ becomes larger, there is higher (lower) frazil concentration in the upper (lower), supercooled (overheated) part of the ISW plume. Owing to the assumption that thermohaline exchanges between frazil crystals and ambient water occur only at the crystal edge for freezing, but over the whole crystal surface for melting (Jenkins and Bombosch, 1995), the integral values of $I_{gr}$ for the lower overheated part can be of much greater magnitude (Fig. 2b). It is therefore necessary to limit the mass loss due to frazil melting in one model time step such that it does not exceed the frazil concentration in the lower, overheated part of the plume. Overall, the frazil concentration and frazil growth rate distributions in the VM model show physically-reasonable and desirable characteristics that are absent from the VU model, and the impacts will be demonstrated by evaluation of the VM model in McMurdo Sound.

## 3 ISW model in McMurdo Sound

The unsteady VM and VU models used in this study are described in detail by Cheng et al. (2017). The governing equations for ISW properties and frazil concentration in both VM and VU models remain as they were in the depth-integrated, two-dimensional ISW plume model developed by Holland and Feltham (2006), except for the different treatments of the specific terms associated with the frazil ice growth rate, described above, in the frazil concentration and potential temperature transport equations of the VM model. Both VM and VU models combine the same commonly-used parameterizations of thermohaline exchanges across the ice–water interfaces, specifically a three-equation formulation (Holland and Jenkins, 1999) for the sea ice base and a two-equation formulation for frazil ice (Galton-Fenzi et al., 2012), with a multiple size–class frazil dynamics model (Smedsrud and Jenkins, 2004), to calculate basal freezing ($f'$) and frazil melting/freezing ($w'$), secondary nucleation ($N'$), and precipitation ($p'$). These processes are summarized in Fig. 3. Rather than repeat all the equations here, we recall some of them and present how we set up our ISW plume models on the McMurdo Sound domain.

The model domain (Fig. 1) is delimited by a 45×40 km rectangle in the *x-y* plane with an ISW outflow from beneath McMurdo Ice Shelf. The base of the sea ice in McMurdo Sound is assumed to be horizontal and rough, owing to the presence of SIPL. The drag coefficient of the ice underside is therefore 6-30 times larger than that typically applied in ice-ocean interaction models (Robinson et al, 2017). The parameterization of the sea ice thermodynamics, the assumption of no entrainment of ambient water into the ISW plume, and the boundary conditions at the ISW outflow follow HU14. The initial thickness of the ISW outflow (indicated by blue arrow in Fig. 1) from underneath McMurdo Ice Shelf is set equal to that of the supercooled layer, i.e., $D = D_{SC}$, and the discharge per unit width is set to 0.02 m$^2$ s$^{-1}$. The addition of both an ambient circulation and tides follow HU14: the former, which represented the only source of momentum in the study of HU14, is assumed to be parallel to the Victoria Land coast, in the negative *y* direction, and to be constant throughout the model domain; the latter is calculated using root-mean square tidal speeds from Padman and Erofeeva (2005). Because ISW persists in McMurdo Sound for at least the 8-9 months of the ice growth season (Robinson et al., 2014), all runs are integrated for 240 days. The model resolution and time step ($\triangle t$) are 1 km and 25 s, respectively. The frazil ice size distribution is represented by 5 crystal size classes, and the transfer processes, induced by frazil freezing and melting, between different size classes are calculated using the scheme proposed by Smedsrud and Jenkins (2004). Sensitivity experiments with more crystal size classes yielded qualitatively similar results. The ice concentration at the ISW outflow is evenly distributed among the classes (Holland and Feltham, 2005, 2006; Smedsrud and Jenkins, 2004).

We treat the frazil ice precipitation rate $p'$ as inverted sedimentation and follow the parameterization of McCave and Swift (1976):

$$p' = w_i C_i \left(1 - \frac{U^2}{U_c^2}\right) \times He \left(1 - \frac{U^2}{U_c^2}\right), \tag{3}$$

where $U_c$ is a critical velocity, above which precipitation cannot occur, determined by Jenkins and Bombosch (1995):

$$U_c^2 = \frac{\theta_i (\rho_0 - \rho_i) g 2 r_e}{\rho_0 C_D},$$

where $\theta_i$ is the Shields criterion, $\rho_0$ and $\rho_i$ are reference seawater and ice densities, respectively, $g$ is gravity, $r_e$ is the equivalent radius of a sphere with the same volume as the frazil disk. The frazil ice rise velocity, $w_i$, is calculated by Morse and Richard (2009):

$$w_i = \begin{cases} 2.025 D_i^{1.621} & if\ D_i \leq 1.27\ mm \\ -0.103 D_i^2 + 4.069 D_i - 2.024 & if\ 1.27 < D_i \leq 7\ mm \end{cases},$$

where $D_i = 2 r_i$ is the diameter of a frazil crystal in mm. The inclusion of the Heaviside function $He$ means that negative precipitation (i.e., erosion of previously deposited frazil ice) is not permitted. Because we have no idea about how cohesive the ice crystals are once they have settled, the estimation of an erosion rate would entail additional uncertainties.

The complex processes after the frazil ice precipitates onto the sea ice base are simplified in our model. In order to calculate SIPL thickness $D_P$ at the n$^{th}$ time interval, we adopt the assumptions of HU14 that solid ice fraction within SIPL in

McMurdo Sound is 0.25 based on the observational estimation from Gough et al. (2012) and that the ice crystals, on average, double in volume after precipitation:

$$D_P = \frac{1}{0.25} \times 2 \times \sum_{k=1}^{n}(p'_k \times \triangle t).$$

It should be noted that the volume change factor is a broad estimate, with almost no supporting evidence in the literature to guide it. Coupling our VM model with a model focusing on the processes associated with platelet ice accretion within the sea ice (Buffo et al., 2018) would be necessary to improve on that rough estimate, but is beyond the scope of the present study.

## 4 Results

### 4.1 Standard model run

The performance of the VU and VM models in reproducing the ISW supercooling and SIPL pattern in McMurdo Sound is evaluated by comparing results with observational data. To our knowledge, the data reported by HU14 are the most comprehensive available to evaluate our model, including both oceanographic and drill-hole measurements in two horizontal dimensions adjacent to McMurdo Ice Shelf. As this study represents the first application of a two-dimensional ISW plume model to the McMurdo Sound region, extensive tuning of the least constrained model parameters, including the ISW outflow properties, SIPL basal drag coefficient, frazil ice crystal size distribution, ambient current speed, and Shields criterion was required to produce the distributions of ISW properties and SIPL thickness shown in Figs. 4a and 6, respectively. Despite the limited observational constraints on many of these parameters we do find support in the literature for our adopted values: ISW outflow properties are consistent with those reported by HU14, and the corresponding thickness of supercooled layer is within the observed range (60-70 m) given in both HU14 and Robinson et al. (2014); the basal drag coefficient fits appropriately within the range identified by Robinson et al. (2017), while the ambient current speed is consistent with the lowest speeds reported in that study; we used 5 crystal size classes, as did Galton-Fenzi et al. (2012), although our sizes are slightly larger; we used a larger Shields criterion than the middle (0.05) of the observed range, although there is considerable scatter amongst the individual results reported from sedimentary experiments. Table 1 summarises all the values adopted for the key parameters. Model results are evaluated by means of skill metrics: Root-Mean-Square Error (RMSE), Correlation Coefficient (CC), and Skill Score (SS), respectively given by

$$RMSE = \left[\frac{\sum(X_{cal}-X_{obs})^2}{M}\right],$$

$$CC = \frac{\sum(X_{cal}-\overline{X_{cal}})(X_{obs}-\overline{X_{obs}})}{[\sum(X_{cal}-\overline{X_{cal}})^2 \sum(X_{obs}-\overline{X_{obs}})^2]^{1/2}},$$

$$SS = 1 - \frac{\sum(X_{cal}-X_{obs})^2}{\sum(X_{obs}-\overline{X_{obs}})^2},$$

where $X$ is the variable being evaluated, $M$ is the number of data points, and the overbar denotes the arithmetic mean. The performance of each model is indicated by SS as: >0.65 excellent; 0.65–0.5 very good; 0.5–0.2 good; <0.2 poor (Luo et al., 2017; Ralston et al., 2010; Song and Wang, 2013).

It can be seen that at the end of the simulations both VM and VU models reproduce the observed reduction in ISW supercooling at the sea ice base ($T_{SC}^0$, superscript "0" denotes the sea ice base) in the cross- and long-sound directions, in spite of some evident model discrepancies (Fig. 4a) that may result from the limitations in our model setup: both the ambient current and tides are treated as temporally and spatially constant; there are no long-term observations of ISW outflow to provide reliable boundary conditions; we use a constant drag coefficient, ignoring the spatiotemporal evolution of the sea ice

basal form characterized by SIPL. We also ignore the impact on ISW properties of brine drainage from the upper SIPL as it is incorporated into the sea ice by the freezing up of interstitial water, driven by heat loss to the atmosphere. Including such processes would require coupling with a sea ice model such as that of Buffo et al. (2018) mentioned above. Nevertheless, the SS of $T_{SC}^0$ calculated using VM and VU models are 0.56 and 0.58, respectively, and the CC and RMSE are also reasonable (Table 2). There are only small differences throughout the time series of $T_{SC}^0$ simulated by the VM and VU models (Fig. 4)

and the final distributions of both total ISW plume thickness and supercooled thickness are also very similar (see Fig. 5a-d). A comprehensive comparison of $T_{SC}^0$ calculated by the VM and VU models in an extensive set of sensitivity experiments will be discussed later. Finally, it can be seen that in both models the ISW plume flow is predominantly governed by a geostrophic balance (Fig 5a-d).

In contrast, the frazil concentration (red lines in Fig. 4b, Fig. 5e and f) and SIPL thickness (green lines in Fig. 4b, Fig. 6b and c) are both underestimated by the VU model, compared with the results of the VM model, throughout the time series. Given the small differences in $T_{SC}^0$ calculated by VM and VU models, this result demonstrates that the vertical distribution of frazil concentration within the ISW plume plays a critical role in determining the suspended frazil ice growth (Fig. 2), and thus the frazil concentration and SIPL thickness distributions. The supercooling is utilized more efficiently in the VM model, giving

a greater depth-averaged frazil concentration than is produced by the commonly-used VU model. The simulated SIPL thickness near the ISW outflow exhibits steeper gradients than are observed (Fig. 6b and c), which probably result from the spatial non-uniformity of ISW plume near the outflow (Fig. 5a and b). That non-uniformity in flow leads to localized non-uniformities in thermodynamics (Fig. 5c and d), frazil concentration (Fig. 5e and f), and thus SIPL thickness (Fig. 6b and c). Moreover, because the sea ice base is horizontal, there are no changes in the freezing point associated with pressure change,

so supercooling is always highest at the ISW outflow (Fig. 5c and d). That results in the greatest frazil concentration (Fig. 5e and f) and SIPL thickness (Fig. 6b and c) near the location of the outflow, and because the outflow is steady in time spatial gradients in SIPL close to the outflow are enhanced. In reality, temporal changes in the ISW outflow position, width, supercooled layer thickness and duration could lead to a broader region of elevated frazil precipitation and a less peaked distribution of SIPL thickness. In addition, such small-scale features in the SIPL thickness distribution, if present, would not

be resolved by the relatively coarse spatial distribution of drill-hole measurements (dots in Fig. 6). Nevertheless, the largest SIPL thickness undoubtedly occurs adjacent to the ISW outflow in McMurdo Sound, and the SIPL thickness calculated by the VM model at drill sites agrees well with the measurements (Fig. 6a), being graded "excellent" in contrast with the "poor" performance of the VU model (Table 2). Despite efforts to tune the VU model to give a better match with the observed SIPL thickness, even a limited expansion of SIPL can only be achieved with a considerable increase in the calculated $T_{SC}^0$, in disagreement with the observations.

For both VM and VU models, the time series of area-averaged $T_{SC}^0$, $C_i$ (hereafter $T_{SC}^0$ and $C_i$ denote their area-average values), and SIPL thickness indicate respectively two near-constant values and one near-constant growth rate after about the 150[th] day (Fig. 4b). It is informative to explore how our various assumptions about the vertical distribution of frazil concentration influence the steady-state relationship between those variables in the McMurdo Sound region.

## 4.2 Dependence of SIPL thickening rate on ISW supercooling

The response of ice shelf basal melting to variations in ocean temperature has been investigated using satellite altimetry (Rignot and Jacobs, 2002; Shepherd et al., 2004) and numerical models (Grosfeld and Sandhäger, 2004; Holland et al., 2008; Payne et al., 2007; Walker and Holland, 2007; Williams et al., 1998, 2002). In contrast, we know of no studies to date that provide a quantitative relationship between marine ice (or SIPL) thickening rate beneath ice shelves (or sea ice) and ISW supercooling. Such a relationship is of potential significance for evaluating the mass balance of deep-draughting ice shelves in cold water environments and adjacent sea ice subject to climatic variability.

Owing to the number of poorly-constrained parameters in the frazil-ice-laden ISW plume model, we conducted 211 comparative sensitivity experiments between VM and VU models, varying both physical and input parameters, including drag coefficient, frazil ice crystal size configuration, average number of frazil crystals, ambient current speed, width and thickness of the ISW outflow, and frazil concentration within the outflow (see Table 3). For all model runs, we plot the relationship between $T_{SC}^0$ and thickening rate in the steady state, using output from the last 30 days of each run (Fig. 7).

In Fig. 7a, the results of the VM model are grouped by the prescribed supercooled layer thickness $D_{SC}^{ini}$ in the ISW outflow. For $D_{SC}^{ini}$ <65 m there is a relatively consistent increase in thickening rate with increasing $T_{SC}^0$, while for $D_{SC}^{ini}$ ≥65 m the thickening rate tends to be much more variable. It is worth mentioning that $D_{SC}^{ini}$ =65 m is the value estimated by HU14 based on the measurements conducted by Lewis and Perkin (1985) and Jones and Hill (2001). For $D_{SC}^{ini}$ =78 m and greater, inflexions emerge separating a region of low thickening rate, where the thickening rate tends to decrease with increasing $T_{SC}^0$, from a region of high thickening rate, where there is a very rapid increase in thickening rate with increasing $T_{SC}^0$. This

complex response of the VM model must result from the consideration of vertical structure in the frazil concentration, controlled by the frazil ice suspension index $Z_*$ (Fig. 2), in the calculation of frazil ice growth.

We therefore calculated the weighted-average of $Z_*$ at each grid point for the VM model using the following equation:

$$\bar{Z}_* = \frac{\sum_{k=1}^{n} C_i^k Z_*^k}{\sum_{k=1}^{n} C_i^k} = \frac{\sum_{k=1}^{n} C_i^k Z_*^k}{C_i},$$

where $C_i^k$ is the frazil concentration of the k[th] size class, and n is the number of size classes used. Then, we took the average of $\bar{Z}_*$ over all the grid points occupied by the plume to give a representative suspension index for the VM runs (hereinafter $\bar{\bar{Z}}_*$ denotes its area-averaged value). We replotted the VM model results characterized by $\bar{\bar{Z}}_*$ in Fig. 7b. We find systematic changes in $\bar{\bar{Z}}_*$ with increasing thickening rate (along the coloured lines in Fig. 7b), particularly for $D_{SC}^{ini} \geq 78$ m where the

10 inflexions emerge. With decreasing $\bar{\bar{Z}}_*$, $T_{SC}^0$ first decreases, and then increases. If $\bar{\bar{Z}}_*$ is sufficiently large, the suspended frazil crystals deposit out of the ISW plume so rapidly that they cannot efficiently use the ISW supercooling to grow, leading to the smallest SIPL production for the VM model. For smaller $\bar{\bar{Z}}_*$, the frazil crystals bathed in the supercooled layer of the ISW plume can remain in suspension and grow longer, resulting in a thicker SIPL and less residual supercooling. However, if $\bar{\bar{Z}}_*$ decreases further, higher frazil concentration occurs within the lower, overheated part of the ISW plume, where melting of

15 the crystals can mitigate the release of latent heat (Fig. 2b). That promotes further growth of frazil ice which can remain in suspension even longer, and thus lead to rapid SIPL production. The thickening rate calculated by the VU model is also shown, and is discernibly smaller than that calculated by the VM model. In addition, the maximum values of $T_{SC}^0$ were obtained within the VU model, because the supercooling is used less efficiently for producing SIPL in the VU than in the corresponding VM runs.

These arguments can be further illustrated by a more detailed comparison of $T_{SC}^0$ calculated by the VM and VU models (Fig. 8). There are a number of runs, including the standard run, that have larger $T_{SC}^0$ values in the VM than in the VU model. The trend from larger $T_{SC}^0$ in the VM model to larger $T_{SC}^0$ in the VU model is accompanied by increases in $\bar{\bar{Z}}_*$. When $\bar{\bar{Z}}_*$ is relatively small, large frazil concentration exists within the lower overheated part of the ISW plume (Fig. 2b) where melting

of frazil ice (causing cooling) counteracts the consumption of supercooling by frazil growth (causing warming) in the upper part of the plume. As $\bar{\bar{Z}}_*$ increases, the frazil concentration within the lower overheated part decreases, and finally vanishes, and the resulting release of supercooling in the upper part is more efficient in the VM model, giving larger $T_{SC}^0$ values in the VU model.

In Fig. 7a, when $D_{SC}^{ini} < 65$ m, ISW supercooling is insufficient to distinguish runs with different $\bar{\bar{Z}}_*$. In other words, the relation between thickening rate and $T_{SC}^0$ is independent of $\bar{\bar{Z}}_*$ for such small $D_{SC}^{ini}$. When $D_{SC}^{ini}$ is within the range of 65 to 78

m, the VM model results are distinguishable, with data points having smaller thickening rate and larger $T_{SC}^0$ corresponding to larger $\bar{Z}_*$ (Fig. 7b). When $D_{SC}^{ini} \geq 78$ m, the inflexions emerge, and the ISW supercooling revives when $\bar{Z}_*$ decreases further. Therefore, we conclude that when $D_{SC}^{ini}$ exceeds a critical value (about 65 m for these McMurdo Sound simulations), the efficiency of converting ISW supercooling into frazil ice growth is controlled by the suspension index.

### 4.3 Dependence of frazil concentration on ISW supercooling

In view of the correlation between SIPL thickening rate and frazil concentration shown in Eq. (3) (also see Fig. 5e and f; Fig. 6b and c), we will explore the relationship between $T_{SC}^0$ and $C_i$ here. As expected, the complex response of $C_i$ to variations in $T_{SC}^0$ (Fig. 9) is similar to the relationship between $T_{SC}^0$ and thickening rate (Fig. 7) in the VM model.

The magnitude of the difference in $C_i$ calculated by VM and VU models (VM minus VU) is compared in Fig. 10a, where we find that $C_i$ calculated by the VM model is always larger than that calculated by the VU model. In general, the difference increases with decreasing $\bar{Z}_*$, while the sensitivity grows with increasing $D_{SC}^{ini}$. The dependence on $\bar{Z}_*$ is once again due to the impact of the combined thermodynamic processes, i.e., the efficient growth in the upper supercooled part of the plume together with the maintenance of supercooling by melting of frazil in the lower part, discussed above. We also see similar behavior for the difference in the thickening rate (Fig. 10b).

Fig. 7 (Fig. 9) suggests a possible relationship between SIPL thickening rate (frazil concentration) and supercooling in McMurdo Sound, but observations of suspended frazil ice crystal sizes and turbulence within the ISW would be needed to calculate a representative suspension index. To date, there are limited observations of frazil ice in situ, and the majority of the observations made use of instruments not specifically designed for ice crystal detection (Leonard et al., 2006).

### 5 Summary and future works

In this study, we demonstrated how the vertical distributions of supercooling and frazil ice concentration within an ISW plume jointly determine the growth of suspended frazil ice, and thus the rate of SIPL formation under sea ice and marine ice beneath ice shelves. A vertically-modified, frazil-ice-laden, ISW plume model which encapsulates these combined nonlinear effects was applied to the McMurdo Sound region, and reproduced the observed ISW supercooling and SIPL distributions in two horizontal dimensions. Using multiple model runs, the relationship of ISW supercooling to SIPL thickening rate and frazil concentration in McMurdo Sound was explored, and shown to be dependent on the suspension index that controls the vertical distribution of frazil concentration within the ISW plume. Moreover, when the thickness of a supercooled layer of ISW is large enough, the efficiency of converting ISW supercooling into frazil concentration, and thus SIPL growth is determined by the suspension index. These findings highlight the need for further observations in McMurdo Sound, particularly focused near the ISW outflow region in the western sound, where the supercooled ISW plume and SIPL are

prominent, and more general observations that help to constrain the frazil size spectrum within the sea ice-ocean boundary layer. In addition, the performance of the VM model in providing reliable estimates of supercooling and frazil ice flux at the SIPL base makes it an attractive tool for coupling with sea ice models focusing on microscale processes within the bottom layer of the ice (Buffo et al., 2018).

It would be straightforward for the next step to investigate the relationship between supercooling and marine ice thickening rate underneath ice shelves using the VM model. Quantifying this relationship would be the key to parameterizing the process in more complex three-dimensional, primitive equation ocean models, which frequently neglect details of the ice shelf-ocean boundary layer and processes associated with an evolving suspension of frazil ice crystals (Liu et al., 2017, 2018; Mueller et al., 2012, 2018). Results may differ from those discussed above, because of the subtly different environments beneath sea ice and ice shelves. Beneath a SIPL, supercooling is produced by the pressure drop experienced by ISW as it emerges from beneath an ice shelf and rises towards the sea surface, while supercooling that drives marine ice accretion beneath ice shelves is produced as the ISW ascends a very gentle basal slope. The in-situ supercooling level beneath ice shelves is therefore likely to be much smaller than that observed in McMurdo Sound, while the differing slopes also yield differing buoyancy forcing on the flow. Furthermore, after experiencing the step-change in pressure as it ascends the ice front, the supercooled plume in McMurdo Sound is in the process of adjustment, through the formation of suspended frazil and direct freezing onto the accreted SIPL, towards an equilibrium that is presumably attained beyond the region of observations. At the base of an ice shelf, typically several hundred meters thick, the vertical temperature gradient is comparatively small, so the deposited crystals form a slushy layer (Engelhardt and Determann, 1987) that slowly consolidates, possibly as much through compaction as freezing. The ice-ocean interface and the associated drag coefficient are therefore likely to be very different to those observed in McMurdo Sound, where SIPL appears to comprise a more open matrix of ice and water that consolidates by freezing as heat is lost to the atmosphere. In addition, the vastly different time scales over which crystal accretion occurs (about 1-3 years in McMurdo Sound vs tens-hundreds of years beneath ice shelves) could lead to further differences in the internal structure of the crystal layers and hence in the physical boundaries they present to the ISW plume. Therefore, the VM model would need to be re-evaluated against observations of sub-ice shelf ISW plumes and the ice shelf-ocean boundary layer. Finally, further process studies, including the influence of the vertical current structure within either the ice shelf or sea ice -ocean boundary layer (Jenkins, 2016; Robinson et al., 2017) could also contribute to improving our understanding of marine ice and SIPL formation.

*Data availability.* The data archive associated with this study can be found in the Global Change Master Directory under the keyword K063_2011_2012_NZ_1.

*Author contributions.* CC led the study. The simulations were designed by ZW and CC, implemented by CL and RX, and analyzed by CC, AJ, and PRH. The paper was written by CC, AJ, and PRH.

*Competing interests.* The authors declare that they have no conflict of interest.

*Acknowledgements.* We would like to thank three anonymous referees and Ken Hughes for their thorough review and helpful comments and suggestions. This work was funded by the National Natural Science Foundation of China (41406214, 41876220, 41306208, 41606217). CC and CL were respectively supported by the China Scholarship Council (201708320046, 201504180026). ZW was supported by "the Fundamental Research Funds for the Central Universities"
(2017B04814, 2017B20714).

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

**Table 1: List of parameters used in standard model run.**

| Parameter | Value | Description |
|---|---|---|
| $f$ | $-1.4244\times10^{-4}$ s$^{-1}$ | Coriolis parameter |
| $D_{ini}\left(D_{SC}^{ini}\right)$ | 78 m | Constant ISW plume outflow thickness (constant outflow supercooled layer thickness) |
| $W_{ini}$ | 3 km | ISW plume outflow width with constant $D_{ini}\left(D_{SC}^{ini}\right)$ |
| $C_i^{ini}$ | $1\times10^{-6}$ | Depth-averaged volumetric frazil concentration in outflow |
| $N_{ice}$ | 5 | Number of frazil ice sizes |
| $r_{i,1}, r_{i,2}, r_{i,3}, r_{i,4}, r_{i,5}$ | 0.2, 0.6, 0.9, 1.2, 1.5 mm | Frazil ice radii for each class |
| $a_r$ | 0.02 | Aspect ratio of frazil discs |
| $\bar{n}$ | $1\times10^{3}$ m$^{-3}$ | Average number of frazil crystals in all size classes per unit volume |
| $C_d$ | 0.02 | SIPL basal drag coefficient |
| $V_a$ | -0.01 m s$^{-1}$ | Ambient flow speed |
| $A_H$ | 100 m$^2$ s$^{-1}$ | Horizontal eddy viscosity |
| $K_H$ | 20 m$^2$ s$^{-1}$ | Horizontal turbulent diffusivity |
| $S_{ini}$ | 34.59 psu | ISW plume outflow salinity |
| $T_{ini}$ | $-0.0573 \times S_{ini} + 0.0832 - 7.61 \times 10^{-4} D_{ini}$ | Potential temperature of ISW plume outflow |
| $\theta_i$ | 0.075 | Shields criterion number |

**Table 2: List of calculated skill metrics for the results of VM and VU standard model runs.**

| Variable | RMSE | | CC | | SS | |
|---|---|---|---|---|---|---|
| | VM | VU | VM | VU | VM | VU |
| $T_{SC}^0$ | 0.0070 ºC | 0.0069 ºC | 0.83 | 0.84 | 0.56 | 0.58 |
| SIPL thickness | 1.034 m | 2.928 m | 0.91 | 0.01 | 0.79 | -0.65 |

**Table 3: Parameter settings for sensitivity runs, indicated by check, colour-coded by ISW outflow thickness (bottom row). All other parameters remain as they were for the standard model run.**

| | | Drag coefficient | | | | |
|---|---|---|---|---|---|---|
| | | 0.015 | 0.0175 | 0.02 | 0.0225 | 0.025 |
| Frazil size configuration (mm) | A: (0.2,0.6,0.9,1.2,1.5) | ✓✓✓✓✓✓✓ | | ✓✓✓✓✓✓ | | ✓✓✓✓✓✓✓ |
| | 1.125×A | | ✓✓✓✓✓✓✓ | | ✓✓✓✓✓✓ | |
| | 1.25×A | | ✓✓✓✓✓✓✓ | | ✓✓✓✓✓✓ | |
| | 1.375×A | | ✓✓✓✓✓✓✓ | | ✓✓✓✓✓✓ | |
| | 1.5×A | ✓✓✓✓✓✓✓ | | ✓✓✓✓✓✓✓ | | ✓✓✓✓✓✓✓ |
| | 1.625×A | | ✓✓✓✓✓✓✓ | | ✓✓✓✓✓✓ | |
| | 1.75×A | | ✓✓✓✓✓✓✓ | | ✓✓✓✓✓✓ | |
| | 1.875×A | | ✓✓✓✓✓✓✓ | | ✓✓✓✓✓✓ | |
| | 2×A | ✓✓✓✓✓✓✓ | | ✓✓✓✓✓✓✓ | | ✓✓✓✓✓✓✓ |
| $W_{ini}$-ISW plume outflow width with constant $D_{ini}(D_{SC}^{ini})$ | 1 km | | | ✓✓✓✓✓✓✓ | | |
| | 5 km | | | ✓✓✓✓✓✓✓ | | |
| $C_i^{ini}$-Depth-averaged volumetric frazil concentration in outflow | 0.2×10$^{-6}$ | | | ✓✓✓✓✓✓✓ | | |
| | 5×10$^{-6}$ | | | ✓✓✓✓✓✓ | | |
| $V_a$-Ambient flow speed | 0 | | | ✓✓✓✓✓✓ | | |
| | -0.02 m s$^{-1}$ | | | ✓✓✓✓✓✓✓ | | |
| $\bar{n}$-Average number of frazil crystals in all size classes per unit volume | 200 m$^{-3}$ | | | ✓✓✓✓✓✓✓ | | |
| | 5000 m$^{-3}$ | | | ✓✓✓✓✓✓ | | |
| $D_{ini}(D_{SC}^{ini})(m)$−Constant ISW plume outflow thickness (constant outflow supercooled layer thickness) | | ✓   ✓   ✓   ✓   ✓   ✓   ✓   ✓ | | | | |
| | | 30  50  65  70  78  95  100  110 | | | | |

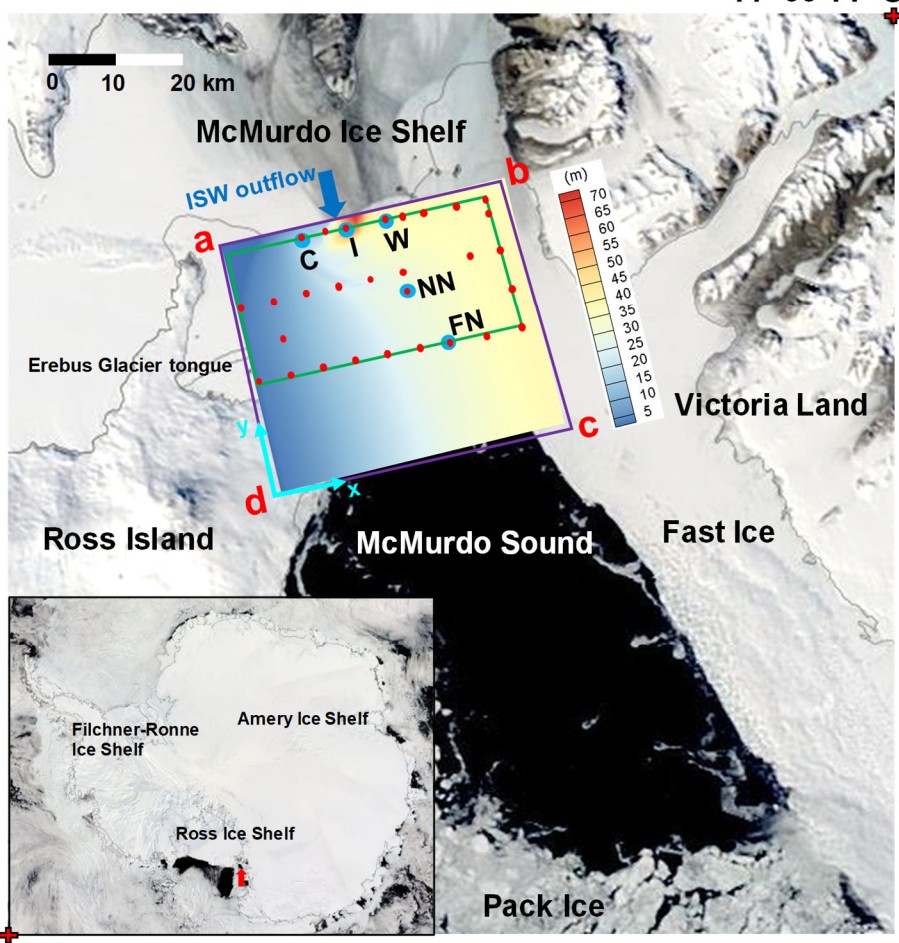

**Figure 1: Satellite image of McMurdo Sound region on 29 Nov. 2011. Purple and green frames outline the model and ice borehole (Fig. 6) domains, respectively. Colours within the purple frame indicate the steady state supercooled ISW plume thickness calculated by the vertically-modified ISW plume model in the standard run (Fig. 5d). Light gray lines outline McMurdo Ice Shelf front and coastlines. Model boundaries d-a, a-b (except the ISW outflow) and "b-c" are treated as solid walls, while "c-d" is an open boundary. Blue and red dots respectively mark the oceanographic CTD and ice drilling sites, and the blue arrow represents the location of the ISW outflow in the model. The red arrow in the inset (bottom-left) points to the location of the McMurdo Sound region. Location names C, I, W, NN, and FN mean Central, Intermediate, West, Near North, and Far North, respectively. Satellite image: NASA Rapid Response MODIS Subsets (http://earthdata.nasa.gov/data/near-real-time-data/rapidresponse/modis-subsets).**

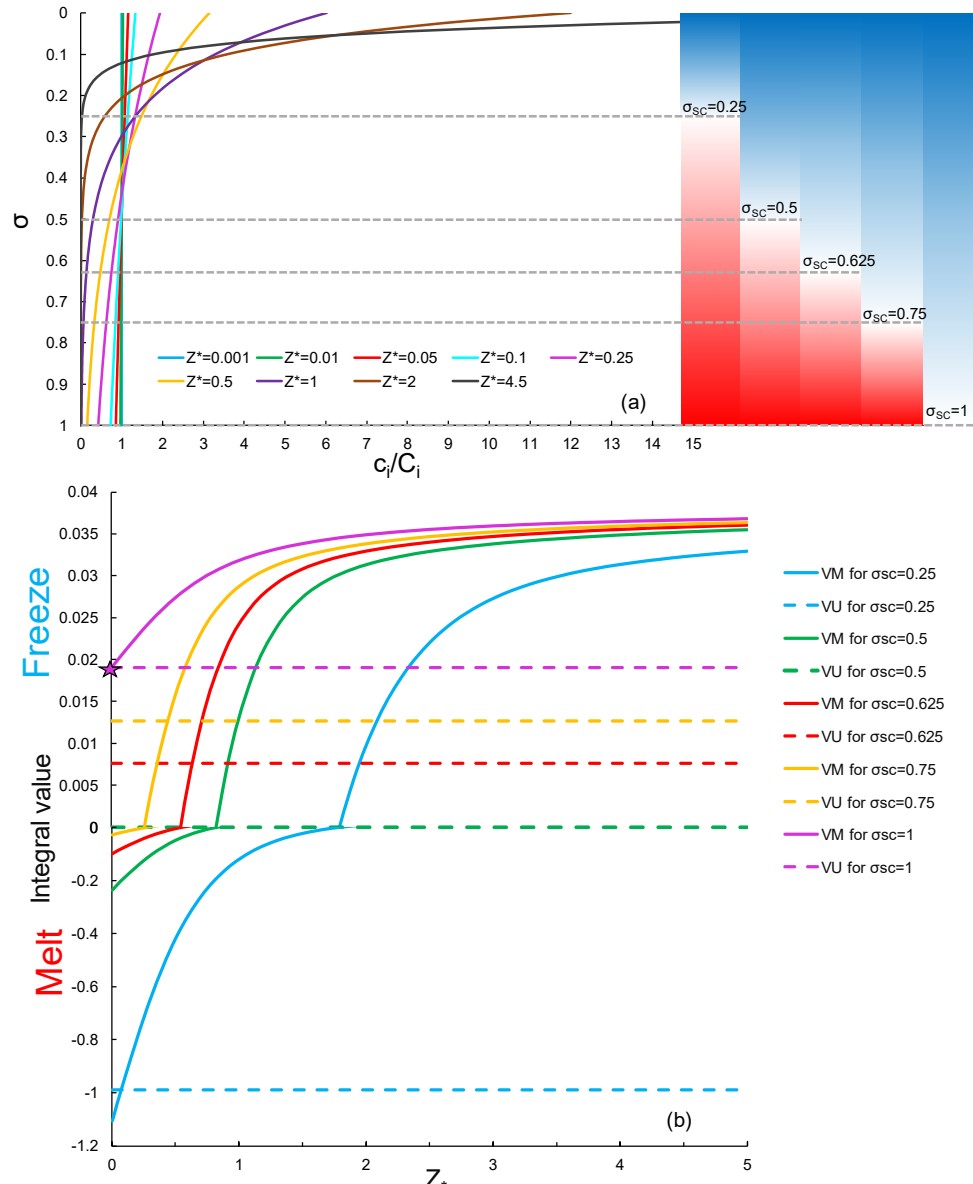

**Figure 2: (a) Exponential profiles of equilibrium frazil concentration for selected values of $Z_*$. Coloured bars at the right and horizontal dashed lines indicate the distribution of supercooling (blue, $T_{SC} > 0$) and overheating (red, $T_{SC} < 0$) for the values of $\sigma_{SC}$ used in (b). (b) Dependence of integral value of $I_{gr}$ on $Z_*$ for suspended frazil ice freezing ($I_{gr} > 0$) and melting ($I_{gr} < 0$) under the supercooling conditions shown in (a). The star denotes the particular conditions under which the integral values of $I_{gr}$ calculated using VU and VM formulations are equal. Note that different y-axis scales are used for freezing and melting.**

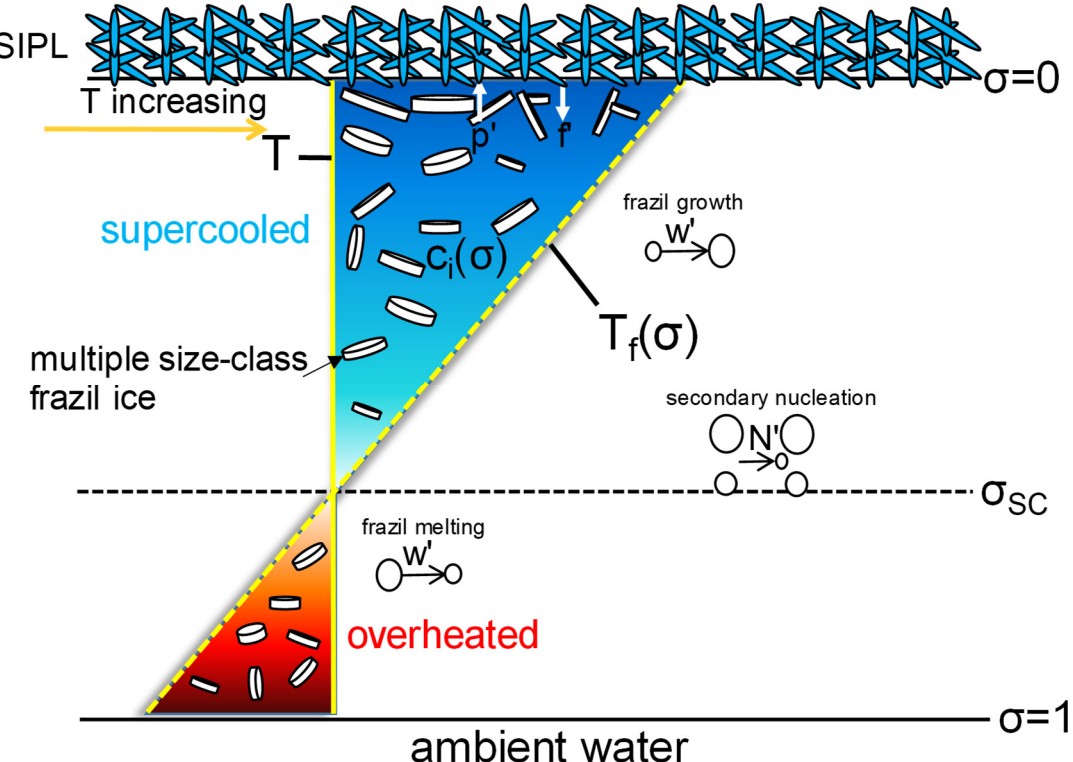

**Figure 3: Schematic diagram of vertical distribution of thermal forcing and relevant processes within a supercooled ISW plume of homogeneous potential temperature and salinity. Secondary nucleation is the process by which the frazil ice in the smallest class is supplemented by collisions between other larger frazil ice crystals.**

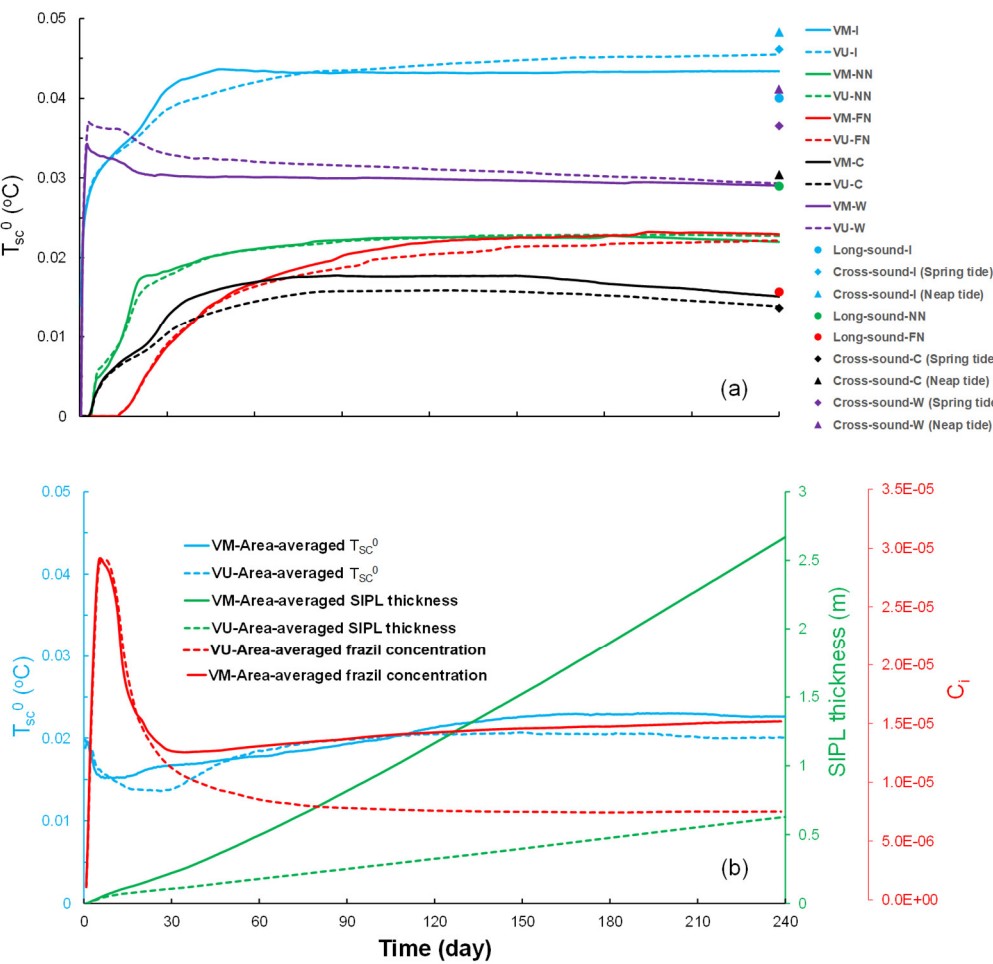

**Figure 4: (a)** Time series of $T_{SC}^0$ simulated by VM (solid lines) and VU (dashed lines) models at five oceanographic sites (colour-coded) in the McMurdo Sound region. **(b)** Time series of area-averaged $T_{SC}^0$ (blue), SIPL thickness (green), and frazil concentration (red) simulated by VM (solid lines) and VU (dashed lines) models over the model domain (purple frame in Fig. 1).

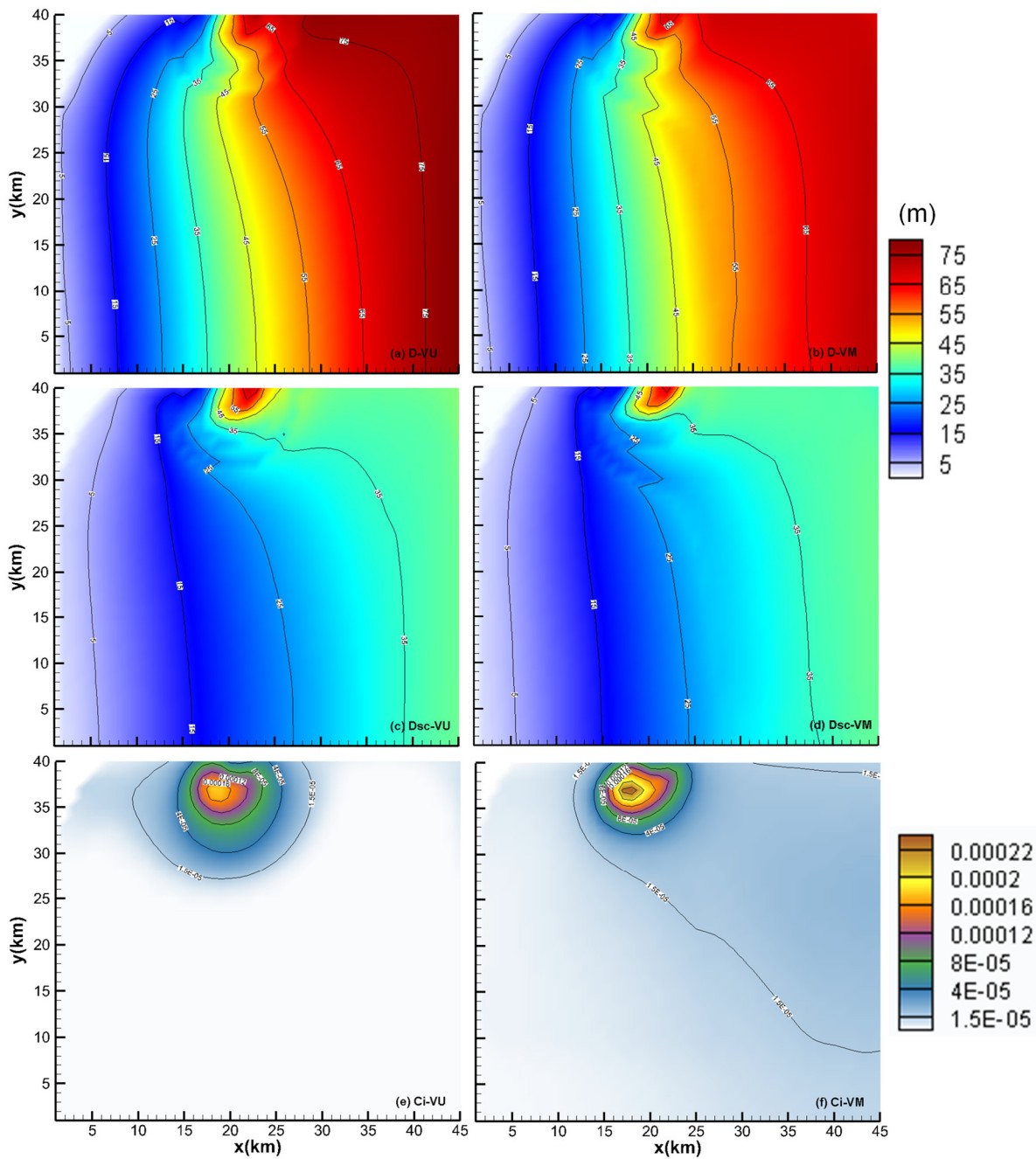

**Figure 5: Spatial patterns, interpolated from model results using Natural Neighbour method, of (a), (b) total, (c), (d) supercooled ISW plume thickness, and (e), (f) depth-averaged frazil concentration at the end of the standard runs of (a), (c), (e) VU and (b), (d), (f) VM models over the domain (purple frame in Fig. 1). Note that the colour scale used in (a-d) is unified.**

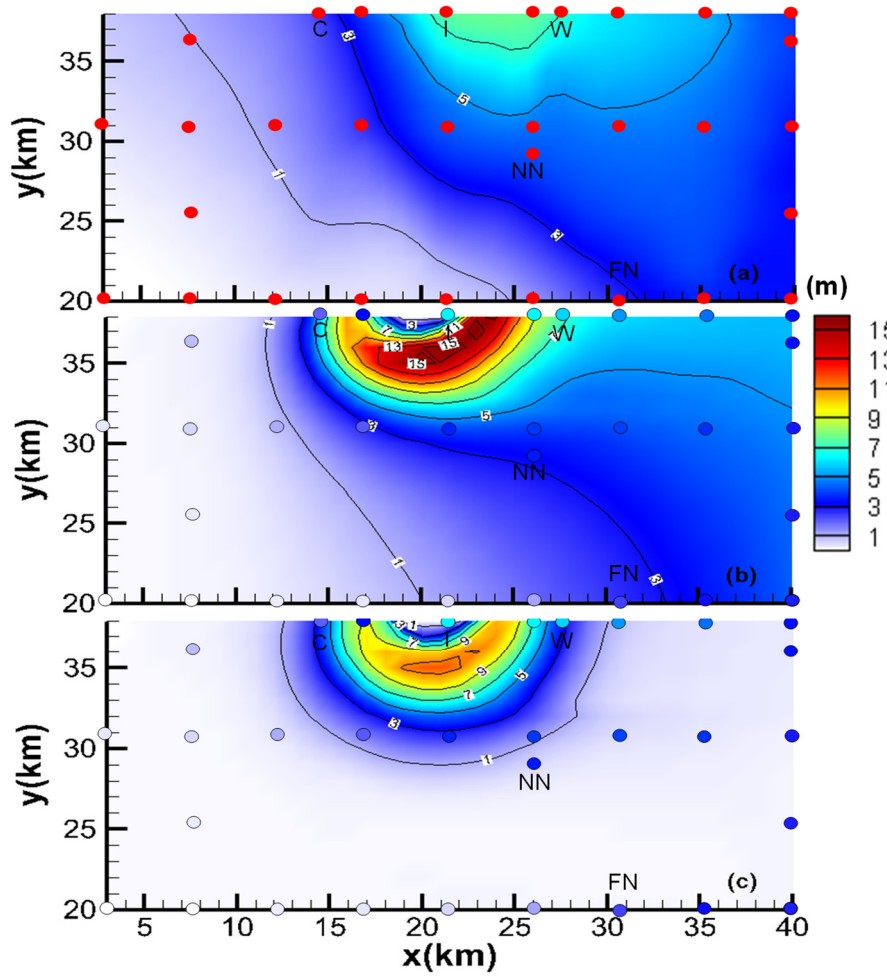

**Figure 6: (a) SIPL thickness over green box in Fig. 1 interpolated, using Natural Neighbor method, from drill-hole measurements (red dots). (b) and (c) SIPL thickness derived from (b) VM and (c) VU models, compared with drill-hole measurements (colour-coded dots). Note that the colour scale is unified.**

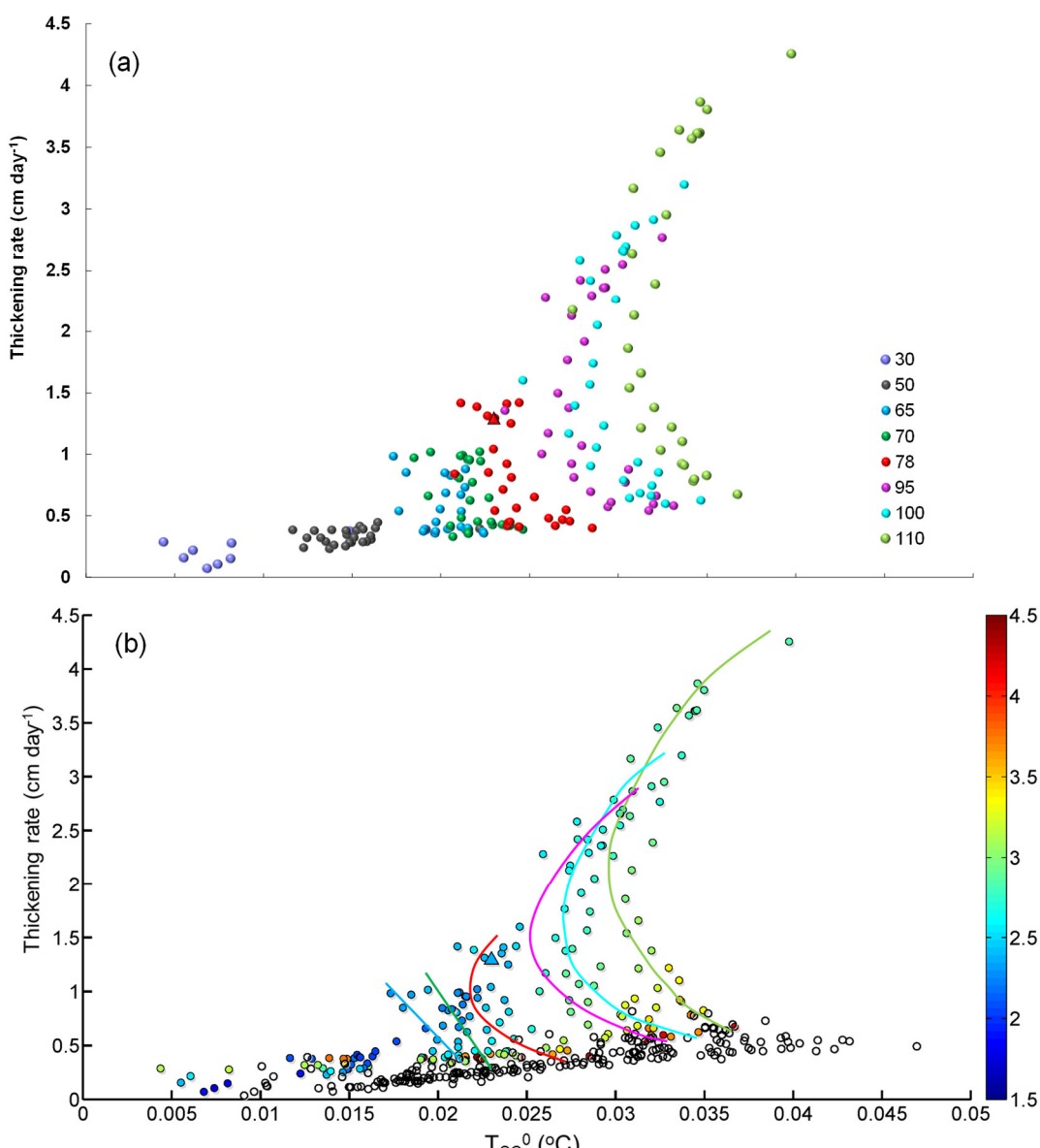

**Figure 7:** Relationship between $T_{SC}^0$ and thickening rate classified by (a) outflow supercooled layer thickness $D_{SC}^{ini}$ and (b) $\overline{Z_*}$ (colour-coded). Numbers in legend of (a) represent the values of $D_{SC}^{ini}$. Solid and hollow dots in (b) correspond to the VM and VU model runs, respectively. Coloured lines depict the central trend of the corresponding data points shown in (a). Triangle corresponds to the standard run. The results are from the last 30 days of the model runs.

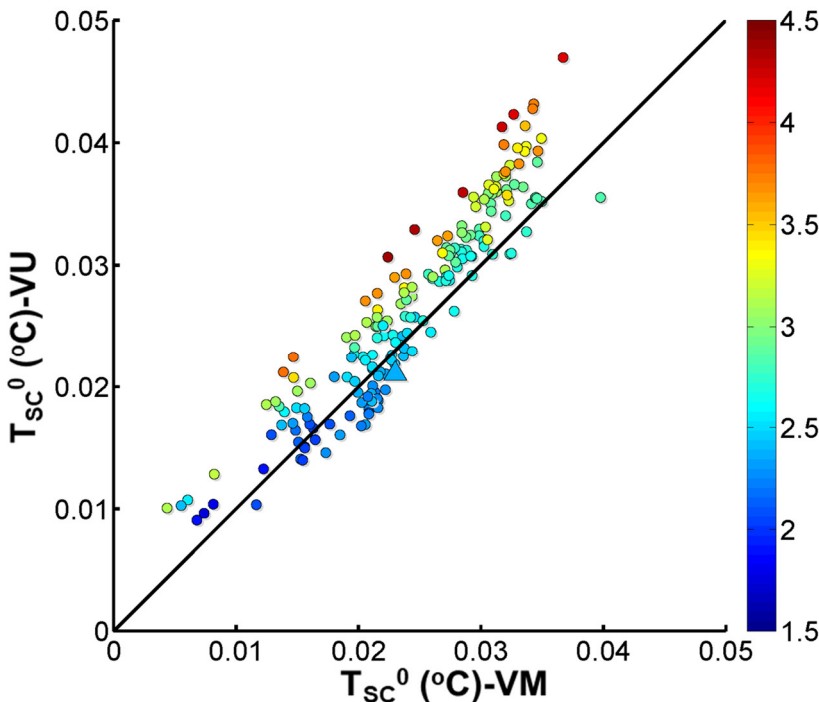

**Figure 8: Comparison of $T_{SC}^0$ calculated by the VM and VU models. Triangle corresponds to the standard run. The colour scale of $\overline{Z_*}$ is the same as in Figure 7b.**

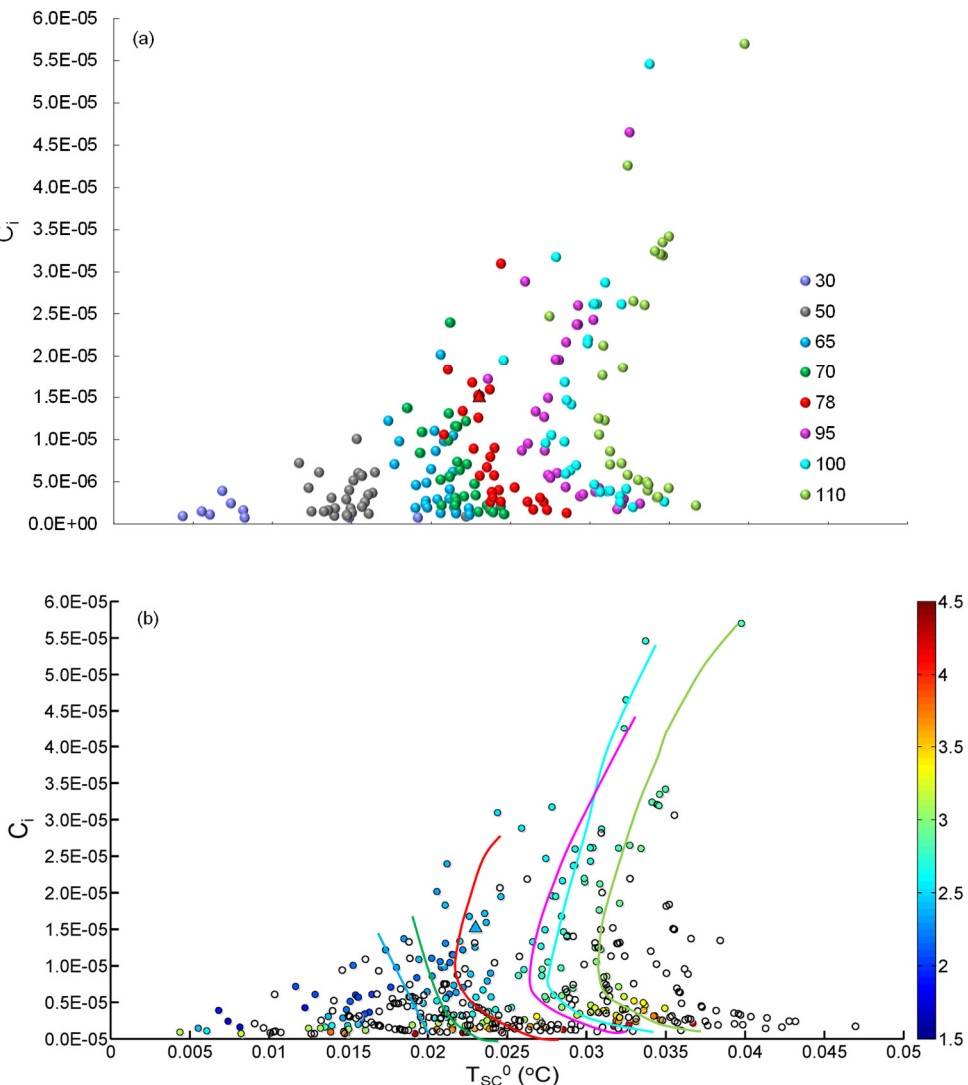

**Figure 9: Same as Fig. 7, but for the relationship between $T_{SC}^0$ and $C_i$.**

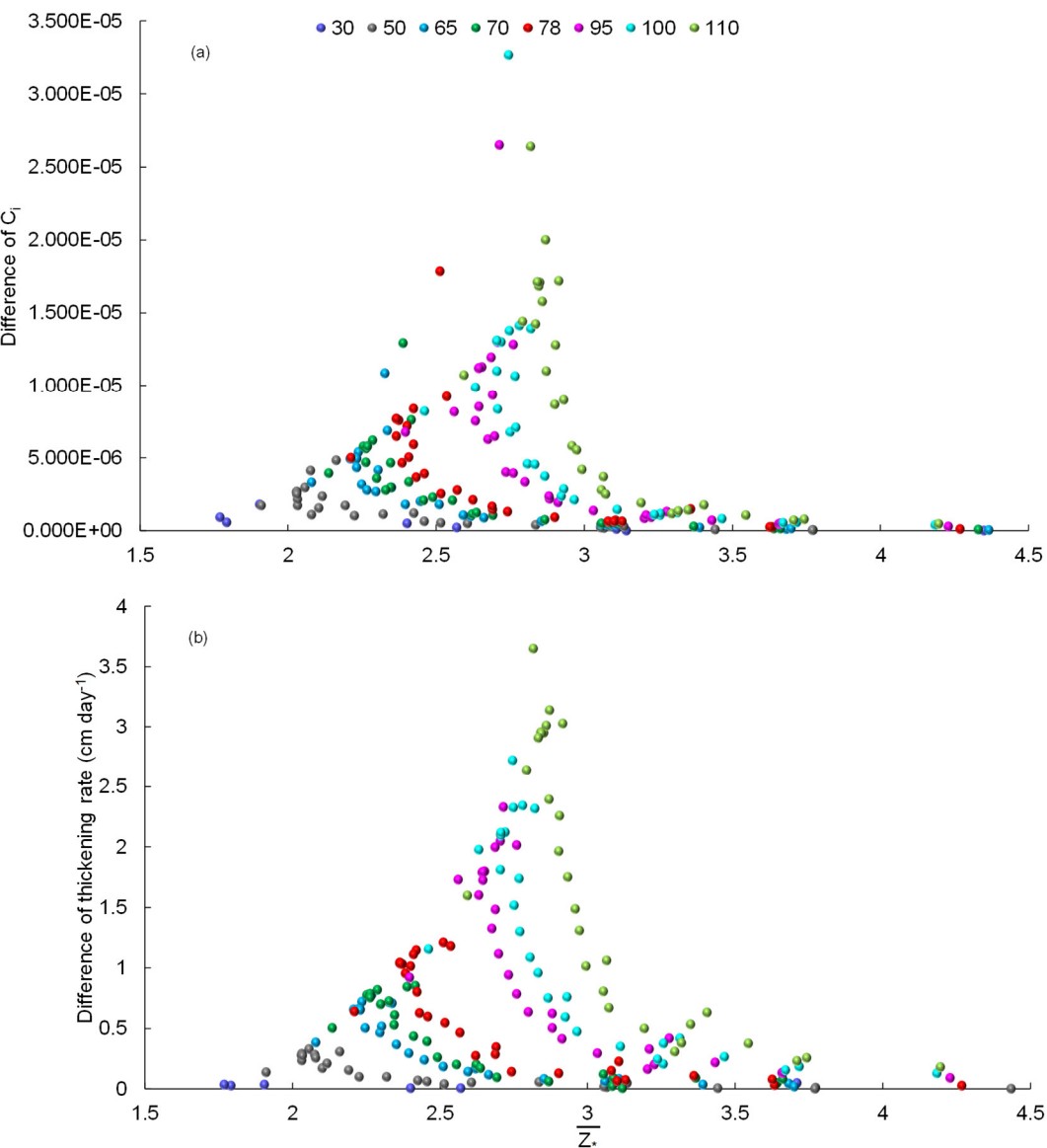

**Figure 10: Relationship between $\overline{Z}_*$ and difference of (a) $C_i$ and (b) thickening rate calculated by VM and VU models (VM minus VU), classified by outflow supercooled layer thickness $D_{SC}^{ini}$. Numbers in legend represent the values of $D_{SC}^{ini}$.**