# Peer review of "Responses of sub-ice platelet layer thickening rate and frazil ice concentration to variations in Ice Shelf Water supercooling in McMurdo Sound, Antarctica"

_The Cryosphere, 2018_

## Referee Comment (RC1) · Anonymous Referee #1 · 29 Aug 2018

**1   General comments**

The goal of the paper is to understand the relationship between supercooling, frazil concentration and platelet ice accumulation rate. The authors apply a model that some of them had published in JPO in 2017. The claimed novelty of this model is that it considers the vertical distribution of supercooling and frazil ice concentration. The model is applied to McMurdo Sound and agrees with observations better than previous models. The authors try to understand the complex behaviour of the model through a (in some ways extensive) series of sensitivity experiments.

My overall evaluation is that the paper needs major revision before it could be published in *The Cryosphere*. I do think that the paper is interesting, timely, and that the idea of improving frazil ice models by considering variation with depth is worthwhile. However, I don't think the modelling performed is particularly novel and there are important limitations to the sensitivity experiments performed that limit the significance of the results. The explanations of the results (which are arguably the main interest for a general audience) are somewhat underdeveloped. I hope the authors are able to address the specific issues, which I discuss in detail below

**2   Specific comments**

1. The authors claim there are two differences in their model compared to previous studies. They consider: (1) the vertical distribution of supercooling; (2) the vertical distribution of frazil ice.

   I think that effect (1) is not really novel, despite the claim 'Cheng et al. (2017) introduced ... the linear depth-dependence of supercooling into [equation] (1)' (see page 4, line 2). As I understand, several previous studies that the authors reference considered this effect. For example, Smedsrud and Jenkins (2004) mention the 'depth-averaged freezing point used to calculate [the growth rate]' in paragraph 32. Jenkins and Bombosch (1995) seem to say the same thing (around equation 15). I could certainly believe that this effect matters, but I don't think this paper can really claim to be novel in this respect, at least in the context of ISW plume models. The discussion in Cheng et al. (2017) seems to better reflect that effect (1) has been previously considered by various studies.

   Effect (2) is novel in this context, although there are some studies in other contexts. For example, see Svensson and Omstedt (1998) in Cold Reg. Sci. Tech. The authors should reference and discuss this the introduction. However, I am

not *entirely* persuaded that effect (2) is important in the context of this study. Naively, I would expect that the frazil rise velocity (say mm/s) is much smaller than the shear velocity, in which case the frazil ice concentration is almost vertically uniform in practice.

2. More broadly, it seems there is the potential for inconsistency in modelling some quantities (such as the water temperature) as vertically uniform, while modelling the frazil ice concentration as varying with depth. Could the authors discuss this issue more clearly and add some justification

3. I think the presentation of the model and the discussion of the results lost sight of the fact that frazil crystals grow by increasing in size.

    In section 2, it was not immediately clear that several quantities like $c_i$ are a function of crystal size, although the authors do point out that crystal size determines the rise velocity.

    Then again on page 8, line 2, the authors analyze the results in terms of some chosen crystal size class that was dominant in some previous studies. Since the model calculated the size distribution (I think), the authors can interpret their results in terms of it. Picking a particular size class is odd because one of the sensitivity experiments involved changing the size classes (i.e. the discretization of the crystal size distribution). Note that the growth rate depends on crystal size, so a simple average size might not be appropriate.

    In the conclusions, page 10, line 2, the authors mention the 'complicated form of the relationship depending on suspension index'. I think this relationship might become clearer through thinking about changes to the crystal size distribution.

4. Equation (2) is somewhat hard to understand. Where does the factor 6 come from? It could be incorporated into $Z_*$ in any case. An advection-diffusion equation in $z$ needs two boundary conditions to determine the two integration constants. One comes from the vertical average $C_i$ but what about the other. It looks

like neither $c_i$ nor its gradient are zero at $\sigma = 0$. Additionally, I think the factor of $\sigma$ in the denominator is a typographic mistake.

[Reading Cheng et al. (2017), it seems that this the factor of 6 is an average inverse diffusivity based on some previous studies, but I think it is important that the present manuscript discusses equation (2) more fully, given that it is the main novel aspect of the paper.]

5. Page 4, around line 15. There is a pair of papers (Rees Jones and Wells, 2015 & 2018, the latter of which you cite) which update the treatment of frazil crystal growth/melting.

    Regarding the need to limit the mass loss due to melting, I think this is a departure from the 'equilibrium' (i.e. steady state) assumption in your vertical ice concentration distribution. Because the local melting rate is proportional to local concentration, the ice mass should just approach zero exponentially over time. Or perhaps there is an issue with your time stepping scheme or use of an excessively large time step (how was 25 seconds chosen)?

6. Sensitivity experiments. In equation (3), there are a couple of 'fudge factors' (solid fraction, volume change) that are not varied. Changing these would be a direct, linear way to get more platelet ice in runs that had less than in observations. Can the authors explain what process the volume change parameter is supposed to represent?

    More broadly, the authors seem to tune a large range of model parameters. This limits the predictive value of the study since these parameters are unknown *a priori*. Could these parameters be used elsewhere or would one need to retune each time? Is the greatly increased drag coefficient (for example) plausible?

    Some of the sensitivity experiments seem very odd. Particularly changing the frazil size configuration (discretization of the crystal equations in size space). If the distribution were well resolved, changing the discretization wouldn't change

the results. I think the author should use more crystals size classes (at least as a test) and should use a smaller minimum crystal size. The minimum size is often very important in these models because it ends up being the size of nucleated crystals. A minimum size of 0.2 mm, therefore, seems large.

7. Section 4. I felt the explanations could have been clearer. For example, phrases like 'supercooling is utilised more efficiently' seem to be key but I didn't understand precise;y what this meant or why it happened.

8. The use of acronyms was excessive for my taste. For example, page 5, line 24 contained five acronyms. I would recommend using more sparingly. Personally, I would change SIPL to 'platelet layer'; FIC to 'frazil concentration'; ASTR to 'thickening rate'. I would change VM and NVM (in any case NVM is an odd name, a double negative, perhaps 'vertically uniform' would be better). In a similar vein, the notation is excessively complicated, with an over-use of 'modifiers' (to give a couple of examples, among many, $\overline{T_{SC}^0}$, $Z_*^a$).

9. The figures contained a lot of information so they are necessarily somewhat complicated. However, all figures are particularly difficult to interpret printed in black and white. In figures 7–10, I would have coloured each point according to $Z_*^a$, rather than dividing into bands. The complicated legends could then be replaced by a simpler colour bar.

**3   Technical comments**

1. Title: I think the paper is broader than just the thickening rate

2. Abstract: I would particularly avoid using so many acronyms in the abstract

3. Abstract, line 16 'choice of frazil ice suspension index': is it really a choice? The meaning of the term 'suspension index' is not clear at this stage.

4. Page 1, line 25: maybe delete 'the' before 'elevated pressure'

5. Page 2, line 14: clause that starts 'in which supercooling' belongs in or immediately after previous sentence.

6. Page 2, line 24: how important are these differences ($D$ not constant, 2 lateral dimensions) in the context of this paper?

7. Page 3, line 5–13: split sentence

8. Page 3, line 6: 'can be quantified through' is vague. Perhaps 'is proportional to'

9. Page 3, line 8: $I_{gr}$ not clearly defined

10. Page 3, line 9: [0,1], use a comma

11. Page 3, line 27: don't understand $U_p + U_a$ or $U_p(U_a)$? Similarly with $V$.

12. Page 3, line 28: italicize $x$ and $y$ (also elsewhere)

13. Page 5, line 7: reads slightly oddly because the 'outflow' is actually the inflow to your domain

14. Page 5, line 16: do you conserve latent heat in this the crystal volume doubling step?

15. Page 7, line 4–8: I suggest that you think about steady state solutions

16. Page 8, line 18: presumably the consumption of supercooling is by the release of latent heat? If so, I would say this explicitly.

17. Page 9, line 13: 'the differences increase with decreasing $Z_*^a$' seemed odd to me, because $Z_*^a = 0$ is supposed to recover the previous models

18. Table 1: some quantities not defined in main text, including $a_r, \overline{n}$

19. Table 1: $T_{ini}$ seems to be a function of $z$, but I thought temperature was vertically averaged in the model?

20. Figure 3: annotations like $p'$ will be impossible to understand unless the reader is very familiar with the frazil literature. I think $w'$ is more usual than $\omega'$ for frazil growth.

---

## Referee Comment (RC2) · Anonymous Referee #2 · 9 Sep 2018

**Summary**

This manuscript identifies, and attempts to quantify, the complexity of the interactions between supercooled water, frazil ice held in turbulent suspension, platelet ice accretion and the temporal evolution of Ice Shelf Water plumes originating in the cold cavities of Antarctic ice shelves. The model used is an evolution of earlier models applied to this problem, with the significant point of difference being inhomogeneous vertical profiles of supercooling and frazil-ice concentration.

In my opinion, the material covered could be of interest to the polar oceanographic community, and its presentation is timely, considering the expansion of interest in the

fate of the Antarctic ice shelves and the sea ice affected by their basal melt. However, I do not believe that the stated 'main objective' has been satisfactorily achieved in light of the limitations of the sensitivity studies. I also think that significant clarification is required before the manuscript would be ready for publication.

Major Points

1. Novelty of approach

a. The introduction of depth-dependent supercooling and frazil ice concentration was explored in a practical setting by Cheng et al., (2017), so it is not clear that there is a great deal of novelty in the present study.

2. Confusion around terms and purpose

a. The general purpose of the study appears to be to improve model representation of marine ice accretion beneath ice shelves. That being the case, some justification is required for evaluating the model against an ISW plume beneath sea ice, which is generating a sub-ice platelet layer (as opposed to accreted marine ice). While there is clearly a lot of connection and similarity between these, some explanation of the differences and limitations of applicability need to be provided. For example: the difference in absolute pressure and its consequence for supercooling potential; the effect of basal slope for generating buoyancy-derived momentum; the likely strength of flow and turbulence in the boundary layer and implications for size of crystal held in suspension.

b. A comment is also required on the general applicability of a model evaluated against a sub-sea ice ISW plume for the ice shelf cavity.

c. No definitions of the terms 'frazil ice' and 'platelet ice' are offered, and direct similarity between sub-ice platelet layers and accreted marine ice is implied. These terms are not interchangeable, and their use here contributes to a general confusion as to the overall purpose of the study. Use of the term ''platelet-like frazil crystals'' (p1 line 28) exemplifies this confusion (for both the authors and readers). Clear definitions of these

terms would help, as would a statement that the plume in McMurdo Sound is being used to evaluate model performance on the basis that it is the most comprehensively-observed plume available, despite its expected differences (which should be clearly identified) from a sub-ice shelf ISW plume.

d. The general confusion of purpose is demonstrated in that the Abstract opens with the ISW plume in McMurdo Sound (I would argue that the abstract should focus on the model rather than its application), while the Introduction is primaritly concerned with the sub-ice shelf regime.

e. I find the comment in the Conclusions (p. 10, lines 9 – 11) that "additional observations of supercooled ISW, SIPL and frazil size spectrum in western McMurdo Sound are needed to constrain the relationships determining marine ice formation beneath ice shelves", rather strange. The recommendation surely should be to observe these parameters beneath ice shelves, if that is indeed the desired outcome?

3. Inadequate explanation of sensitivity studies and choice of values for parameters

a. Page 5, Line 26-27 justifies the tuning of parameters to reproduce the observed SIPL and supercooled ISW thicknesses, but we are not given an indication of the strength of this tuning. How do the chosen parameters for the standard run compare with the observations, and observed range of values?

b. I would expect that the sensitivity of the model could be just as significant for ar, Nice and solid volume fraction as for the parameters that were included in the sensitivity study. Further testing of the sensitivity to these parameters would be ideal, but at least a statement acknowledging (and justifying) their exclusion is required.

c. The role of settling dynamics is clearly significant here (as in sediment studies), but this is not referred to at all.

Minor Points

The thickness of the accreted platelet layer is a critical parameter for testing the performance of the model. However, the mechanism employed for transferring crystals out of the suspended frazil and into the accreted platelet ice layer is not made explicit.

The manuscript is rather heavy-handed in its use of acronyms:

'ISW' is well-known and can be used freely;

The reader may need a few reminders of what 'SIPL' represents;

Please expand 'MMS' to McMurdo Sound in all cases;

'FIC' should not be used, especially as it is variously interchanged with 'ci';

'VM' is OK;

but 'NVM' is non-intuitive;

'ASTR' is non-intuitive – is it possible to avoid this altogether?

The superscript '0' (as in T0SC) is not explained and left to the reader to figure out.

P 1, Line 27: "a necessary condition for ice crystals to form. . ." either needs a reference or should be removed, as primary nucleation is not thought possible under observed supercooling conditions.

P 2, Line 5-6: "the SIPL should not be ignored. . .". This manuscript is not offering new evidence on the role of the SIPL, so is not the place to be offering this recommendation.

P2, Line 8-9: "Owing to the paucity of direct observations. . ." And yet this manuscript relies heavily on observations collected in McMurdo Sound. I do not feel that the observations available (beyond HU14) are sufficiently acknowledged or applied.

P2, Line 16-17: There is no marine ice production in McMurdo Sound (this is an example of the general confusion identified above).

P2, Line 25-26: here it is inferred that the ISW plume thickness is allowed to evolve in the present study, but this is not stated explicitly.

P3, Line 5: The statement that "the concentration of suspended frazil ice controls the dynamic and thermodynamic evolution of ISW outflows" requires a reference.

I see that a crystal size distribution, incorporating 5 size classes, is being used. But how this is incorporated is not explicitly stated.

a. P 5, Line 13: I am not convinced that the frazil ice crystals should be evenly distributed among size classes. Naively, I would expect a much greater proportion to reside in the smallest size class(es). At the very least, this treatment needs a reference.

b. Is the smallest size class small enough? (I'm not sure on this). Is there a reference to support this?

c. I cannot see anywhere that explicitly states whether crystals move between size classes as they grow/melt; or if and how they are removed from the largest size class to accrete as part of the SIPL.

d. I think more justification is needed on the collapsing of a range of size classes to a single size class for comparative purposes.

Which temperature is being used? (i.e. in-situ, potential, conservative...) In observations of the ISW plume in question (e.g. Dempsey et al., 2010; Robinson et al, 2014; which the authors make reference to), the upper ocean is homogeneous in potential temperature.

I presume that latent heat is being added as ice crystals grow, since the supercooling is variously referred to as being 'released', 'utilized', 'converted' and 'varied'. However, how this is incorporated is not identified or referred to anywhere.

P3, Line 27: To aid clarification, could the word 'background' be changed for 'ambient' (current), since this will align with the subscript used in the symbol.

P3, Line 28: I take from this that the 'tidal' speed is not varying in time, but is effectively
just an additional source of kinetic energy?

P 4, Line 8: Please identify where the use of DSC = 50 m is derived from. Is there observational support for this? Or has it come from HU14?

P 4, Line 13-14: I am confused as to why you would want to avoid frazil melting in the lower part of the plume if it is overheated.

P4, Line 19-20: What (observational?) support is there for determining that the VM model shows 'physically-reasonable and desirable characteristics'? What determines this? Is there a reference to support this statement?

P 5, Eqn (3): the fact that individual ice crystals may double in volume after precipitation shouldn't affect the thickness of the SIPL, since presumably their growth is into the interstitial spaces, and shouldn't cause the entire SIPL to expand?

P5, Line 8: My understanding is that the 'background circulation' applied in HU14 was to provide the buoyancy-driven momentum that the sub-IS plume would naturally possess, but which cannot arise automatically within a model beneath a horizontal ice base (i.e. sea ice). That being the case, this represents a departure from sub-Ice Shelf applications, and should be explained again here.

P 6, Line 10-11: I am not satisfied that there is sufficient support for the statement that 'both models reproduce the observed values of ISW supercooling reasonably well'. Looking at Figure 4a,

a. The structural difference between sites FN and NN is not captured at all;

b. The neap-tide supercooling at site C is very different to the modelled result, and the difference between spring and neap at that site is not reflected in the model timeseries;

c. The observed values at Site W do not coincide with the model timeseries values;

d. The only site that has 'reasonable' agreement is Site I, which is very close to the inflow point of the prescribed ISW plume.

Also, the spatial resolution in the figures, especially of the ISW plume structure and SIPL thickness (figures 5 and 6) seem to greatly exceed the stated resolution of the model (1 km, P 5, Line 12). This then has implications for the statements made about how these are resolved by the model (as in P 6, Line 29). Also, this seems like fairly low resolution for the ISW plume itself, at 3 km wide.

P 7, Line 27ff: Behavior around the 'inflexion' points is particularly interesting. I think it would greatly strengthen the paper if the inflexion points could be investigated in higher resolution (especially in Z* and $\sigma$SC) to the point where the mechanism causing them could be adequately explained. There is an informal explanation offered in lines 28 – 30, but studying this in greater detail could be very useful for determining if this is a model artifact or a real phenomenon that could invite observational investigation.

P 8, lines 9-10: perhaps the wording could be more straightforward? Suggest using "... because the supercooling is used less efficiently for producing SIPL in the NVM than in the corresponding VM runs."

P 8/9 How do the critical thicknesses of the supercooled portion of the ISW plume (65 and 78 m) compare with observed thicknesses of the McMurdo Sound ISW plume?

P 9 Line 8: The sentence is incomplete? Perhaps "... there are analogous inflexion points..."?

P 9 Line 24: The sentence is incomplete? "... that have been taken instruments not specifically designed..."

P10 Line 14-15: Robinson et al. (2017) is an example of such a process study and could be referenced here.

Table 1: Is there support in the literature for choosing ar = 0.02?

Table 3: Could the names of the variables be repeated here so that the table is self-contained?

Figure 1: Why does the model data only extend as far as the borehole domain, and not to the edges of the model domain?

Figure 1: Do the location names (C, I, W, NN, FN) mean anything in the present study? If yes, can these be provided? If no, could they be simplified to A -> E without loss of generality?

Figure 1 caption: The 'oceanographic and SIPL data' are not shown in the figure, so reference to them should be removed from the caption. Unless what is meant are the locations of the oceanographic and SIPL data?

Figure 2b: The almost-vertical blue line to the right, and horizontal blue line along the x-axis appear to be mistakes?

Figure 2b: It would be helpful to add the words 'Melt' and 'Freeze' alongside the appropriate parts of the y-axis. Also, I think that the change of vertical scale is not necessary (and possibly only adds confusion)

Figure 3 caption: "... relevant processes within a supercooled ISW plume of homogeneous temperature and salinity".

Figure 6: Just a suggestion: it would aid comparison if the location dots could be colored by the measured SIPL thickness.

Figure 7a: There is no explicit specification of what 'VM30' through 'VM110' refers to.

Figure 7a and 7b: Text says that the ASTR estimate is from the last 30 days of the model runs. This needs to be repeated here.

Also, converting the ASTR to cm/day may be a more useful unit (and less misleading, given that it only refers to 30 days' worth of growth)

Figure 9: The formulae for the fitted lines could be moved out to the right so as not to become lost amongst the data points.

Figure 10: I don't think the additional enlargements provide any further useful information and could be removed.

Figure 10: Numbers on axes are too small.

---

## Short Comment (SC1) · 9 Sep 2018

I (Ken Hughes) was lead author of the paper on which a number of comparisons are made in the current manuscript. Consequently, I have a unique perspective on this paper. I have included a number of comments that would improve the clarity of the manuscript, which I read with interest. I am not, however, an official reviewer, and so am providing these comments for the authors' reference.

The authors modify an existing two-dimensional, time integrated ISW plume model to (i) test its ability to recreate observed supercooling and SIPL thickness observations and (ii) via a rigorous sensitivity study, investigate which parameters have disproportionate

influence on the spatial distributions.

The idea behind the study is worthwhile as McMurdo Sound provides arguably the best location against which an ISW plume model can be test. Indeed, the two-dimensional model is a noticeable improvement over the 1D approach used by Hughes et al. (2014).

Major issues

Line 5.27 "some tuning of model parameters" overlooks some key details

1. Why is the outflow only 3km? Observations in Hughes et al have ISW outflow adjacent to the ice shelf across the whole Sound. 2. There is no mention that frazil ice classes in this paper are well below the mode size predicted by Hughes et al. (2014) of 2.5mm radius

Line 2.25 Somewhere in the paper, a description of the plume dynamics beneath the sea ice is warranted. Hughes et al. (2014) used a constant plume thickness out of necessity, because one-dimensional models can't easily deal with a lack of forcing (ie a gravitational component beneath a slope). It appears from Figure 1 that the plume in this paper tends toward geostrophic balance. Is this true? Put another way, does having two spatial dimensions and the time dimension avoid the issues faced by Hughes et al.

Line 6.29 Although the overall spatial distribution of SIPL may be graded as "excellent", it predicts a large areas where the thickness is 15m, twice the observations. Similarly, the "poorly" graded NVM model is closer to observations near the outflow. In other words, comparison between the models in terms of SIPL is not a simple as calling one "excellent" and the other "poor".

There is no citation, let alone discussion, of Holland and Feltham (2005, doi:10.1017/S002211200400285X), which specifically deals with the issue of the vertical distribution of frazil ice.

On a similar note, there appears to be no real attempt to constrain Z*, the suspension index. A large range is proposed, but the reader is not given any idea what a reasonable value is.

Minor issues

Line 1.13: this is not a new frazil-ice-laden ISW plume model, but rather an existing ISW plume model with an improved frazil-ice treatment.

Line 1.15: This sentence should written as the dependence of supercooling in MMS on the SIPL thickening rate, not the other way as it is currently. In other words, the current sentence says the dependence of x on y is investigated, yet two sentences later, it says x can be expressed as a function of y.

Alternatively, reword to something like "Using this model, we test how ISW supercooling in MMS influences SIPL thickening rate".

This issue also arises for the heading of Section 4.2

Line 2.10 A reference to Galton-Fenzi et al. (2012) is warranted in this paragraph as they develop a three-dimensional model with frazil ice dynamics. It is not a plume model, per se, but it is does involve a a non-vertically-uniform approach to frazil ice dynamics.

Line 2.16 There is a slight issue with using MMS as an example for areas where marine ice production should be reassessed as it is no longer termed marince ice beneath sea ice.

Line 3.16 Hughes et al. (2014) improved upon the method of using $T\_sc$ at mid-depth. This is noted by Cheng et al. (2017) but is not noted here.

Line 4.30 Standard notation for ISW plume models has basal freezing as m' and frazil ice growth as f'. Is there a reason that here basal freezing is f' and frazil ice growth is omega'?

Line 7.11 Describe the type of observations and model results. Something like "...

investigated with satellite altimetry and two-dimensional model results." (wih the actual adjectives based on the respective, cited papers)

---

## Author Comment (AC1) · 19 Oct 2018

Please see the attached zip file which contains: 1) A pdf file (Response to referees_tc-2018-135.pdf) with the referees' comments, the authors' detailed responses, and extracts detailing changes made in the revised manuscript. 2) A pdf file (Tracked changes_tc-2018-135.pdf), showing changes between the revised manuscript and the original submitted manuscript.

Please also note the supplement to this comment:
https://www.the-cryosphere-discuss.net/tc-2018-135/tc-2018-135-AC1-supplement.zip

---

## Author Response (AR1)

**Response to referees:**
**Response of sub-ice platelet layer thickening rate to variations in Ice Shelf Water supercooling in McMurdo Sound, Antarctica**

5  Chen Cheng, Adrian Jenkins, Paul R. Holland, Zhaomin Wang, Chengyan Liu, and Ruibin Xia

*This response comprises Reply to Referee #1, Reply to Referee #2, and Reply to Ken Hughes. Each of our replies to the referees is structured as a sequence of comments from the referee, our response, and changes made to the manuscript.*

Note: Italic denotes the referees' comments, and the following is our response. "P*L#" denotes line # in page * of the 'tracked changes' version of the manuscript in which blue characters exactly represent the revised part. Double quote represents the excerpt of the revised manuscript. Almost all the references mentioned in this report can be found in the **References** section of the manuscript; otherwise, we have given the full citation.

**Reply to Referee #1:**

*1 General comments*
*The goal of the paper is to understand the relationship between supercooling, frazil concentration and platelet ice*

5 *accumulation rate. The authors apply a model that some of them had published in JPO in 2017. The claimed novelty of this model is that it considers the vertical distribution of supercooling and frazil ice concentration. The model is applied to McMurdo Sound and agrees with observations better than previous models. The authors try to understand the complex behaviour of the model through a (in some ways extensive) series of sensitivity experiments.*

*My overall evaluation is that the paper needs major revision before it could be published in The Cryosphere. I do think that*

10 *the paper is interesting, timely, and that the idea of improving frazil ice models by considering variation with depth is worthwhile. However, I don't think the modelling performed is particularly novel and there are important limitations to the sensitivity experiments performed that limit the significance of the results. The explanations of the results (which are arguably the main interest for a general audience) are somewhat underdeveloped. I hope the authors are able to address the specific issues, which I discuss in detail below.*

We thank the Referee #1 for his/her pertinent assessment of our paper and very constructive comments, which have helped us to improve our manuscript. We hope to address all the specific issues below.

*2 Specific comments*

20 *1. The authors claim there are two differences in their model compared to previous studies. They consider: (1) the vertical distribution of supercooling; (2) the vertical distribution of frazil ice. I think that effect (1) is not really novel, despite the claim 'Cheng et al. (2017) introduced ... the linear depth-dependence of supercooling into [equation] (1)' (see page 4, line 2). As I understand, several previous studies that the authors reference considered this effect. For example, Smedsrud and Jenkins (2004) mention the 'depth-averaged freezing point used to calculate [the growth rate]' in paragraph 32. Jenkins and*

25 *Bombosch (1995) seem to say the same thing (around equation 15). I could certainly believe that this effect matters, but I don't think this paper can really claim to be novel in this respect, at least in the context of ISW plume models. The discussion in Cheng et al. (2017) seems to better reflect that effect (1) has been previously considered by various studies.*

Indeed, previous studies have introduced the linear distribution of supercooling into equation 1 but with the assumption of

30 vertically-uniform frazil concentration, which leads to the adoption of supercooling at the mid-depth of Ice Shelf Water (ISW) plume. However, in this study, we didn't aim to emphasize the vertical distributions of supercooling and frazil concentration separately but their combined nonlinear effects: "A vertically-modified frazil-ice-laden ISW plume model that encapsulates the combined nonlinear effects of the vertical distributions of supercooling and frazil concentration…" (P1L14-15), "Here we first analyze the combined nonlinear effects of the vertical distributions of supercooling and frazil

35 concentration on…" (P3L6), "In this study, we demonstrated how the vertical distributions of supercooling and frazil ice

concentration within an ISW plume jointly determine the growth of suspended frazil ice,…" (P11L2-3). The complex behaviour shown in the results (which have been improved in the revised manuscript) of the sensitivity experiments do reflect these combined effects rather than their independent roles. The previous wording implied some misunderstanding of the novelty, so we have rewritten the sentences by deleting the phrases about the introduction of the vertical distribution of

5 supercooling: ", Cheng et al. (2017) showed that adopting an approach in which the frazil ice growth is calculated using a vertically-uniform frazil concentration results in…" (P2L19-20), "In earlier ISW plume models, because $c_i$ is treated as vertically-uniform, the integral of (1) can be represented by the product of the depth-averaged values $T_{SC}^{0.5}$ (0.5 means at mid-depth) and $C_i$." (P3L27-28), and "Accordingly, Cheng et al. (2017) introduced (2) into (1), …" (P4L25).

10 *Effect (2) is novel in this context, although there are some studies in other contexts. For example, see Svensson and Omstedt (1998) in Cold Reg. Sci. Tech. The authors should reference and discuss this the introduction. However, I am not entirely persuaded that effect (2) is important in the context of this study. Naively, I would expect that the frazil rise velocity (say mm/s) is much smaller than the shear velocity, in which case the frazil ice concentration is almost vertically uniform in practice.*

We have added statements about the work of Svensson and Omstedt (1998) in Cold Reg. Sci. Tech. as well as Holland and Feltham (2005) in J. Fluid Mech. in the **Introduction**. That is, "Idealized one dimensional models confirm that the vertical distribution of frazil concentration cannot remain well-mixed in the upper layers of the ocean (Svensson and Omstedt, 1998) and beneath ice shelves (Holland and Feltham, 2005)." (P2L21-23). Again, we emphasize the combined effects of the

20 vertical distributions of supercooling and frazil concentration in this study, and the results of the sensitivity runs serve to demonstrate its importance. The buoyant rise of the crystals must lead to some deviation from vertical uniformity, and we show the impact of that through the dependence of our results on the suspension index. Our standard run, which matches best with observation, has mean suspension index of 2.36, giving considerable variation in ice concentration over depth (Figure 2).

*2. More broadly, it seems there is the potential for inconsistency in modelling some quantities (such as the water temperature) as vertically uniform, while modelling the frazil ice concentration as varying with depth. Could the authors discuss this issue more clearly and add some justification*

30 It is a commonly used assumption in one- and two-dimensional depth-integrated ISW plume modeling that the water temperature and salinity within the plume are vertically uniform, and that assumption has some support in field observations, particularly those made in zones of freezing. Referring to this issue, we extended the previous sentences to "To date, all the ISW plume models mentioned above have been depth-integrated, and all the scalar quantities, i.e., water temperature, salinity, and frazil concentration in those models are treated as vertically-uniform. The well-mixed temperature and salinity

have been validated by borehole observations beneath the Amery Ice Shelf (Herraiz-Borreguero et al., 2013) and under the sea ice in McMurdo Sound (Robinson et al., 2014; HU14)." in P2L14-17.

*3. I think the presentation of the model and the discussion of the results lost sight of the fact that frazil crystals grow by increasing in size.*

We agree with the fact that frazil crystals grow by increasing in size naturally. In this study, we adopted a commonly used scheme proposed by Smedsrud and Jenkins (2004) to calculate the transfer processes, induced by frazil freezing and melting, between different size classes. To make this fact clearer, we added a statement "The frazil ice size distribution is represented by 5 crystal size classes, and the transfer processes, induced by frazil freezing and melting, between different size classes are calculated using the scheme proposed by Smedsrud and Jenkins (2004)." to the model description in P6L9-11.

*In section 2, it was not immediately clear that several quantities like ci are a function of crystal size, although the authors do point out that crystal size determines the rise velocity.*

*4. Equation (2) is somewhat hard to understand. Where does the factor 6 come from? It could be incorporated into Z∗ in any case. An advection-diffusion equation in z needs two boundary conditions to determine the two integration constants. One comes from the vertical average Ci but what about the other. It looks like neither ci nor its gradient are zero at σ = 0. Additionally, I think the factor of σ in the denominator is a typographic mistake.*

*[Reading Cheng et al. (2017), it seems that this the factor of 6 is an average inverse diffusivity based on some previous studies, but I think it is important that the present manuscript discusses equation (2) more fully, given that it is the main novel aspect of the paper.]*

Thank you for the careful checking of the equations. As suggested, we have given a fuller presentation of the derivation of equation (2):

"

Considering only the balance between the buoyant-rise-induced vertical advection and turbulent diffusion terms, the governing equation for frazil concentration can be written as

$$\frac{d}{d\sigma}\frac{K}{D}\frac{dc_i}{d\sigma} + w_i\frac{dc_i}{d\sigma} = 0,$$

where $w_i$ is the frazil ice rise velocity, determined by ice crystal size, $K$ is the vertical frazil concentration diffusion coefficient, which can be parameterized as vertically constant (Cheng et al., 2013, 2016):

$$K = \frac{1}{6}\kappa u_* D,$$

where $\kappa = 0.4$ is von Karman's constant, $u_* = \sqrt{C_d}U$ is the friction velocity, related to the turbulent intensity within the ISW plume, $C_d$ is the basal drag coefficient, $U = \sqrt{\left(U_p + U_a\right)^2 + \left(V_p + V_a\right)^2 + U_t^2}$ is the total flow speed, $U_p(U_a)$ and $V_p(V_a)$ are the depth-averaged ISW plume (ambient current) speed in the $x$ and $y$ directions respectively, $U_t$ is the root-mean square tidal speed. Using a zero net flux condition in the equilibrium state at the lower boundary of the plume, i.e.,

5  $\frac{K}{D}\frac{dc_i}{d\sigma} + w_i c_i = 0$, for $\sigma$=1

and a Dirichlet boundary condition at the upper boundary, i.e.,

$c_i = c_{i,b}$, for $\sigma$=0

where $c_{i,b}$ is the frazil concentration at the ice shelf/sea ice base, the vertical exponential profile for the equilibrium frazil concentration can be readily obtained (Cheng et al., 2017):

10  $\frac{c_i(\sigma)}{c_{i,b}} = exp(-6Z_*)$,

where $Z_* = w_i/\kappa u_*$ is the suspension index, otherwise known as the Rouse number. Integrating this exponential profile from $\sigma$=0 to 1, we finally obtain the relation between $c_i(\sigma)$ and $C_i$ as

$$\frac{c_i(\sigma)}{C_i} = \frac{6Z_* exp(-6Z_*\sigma)}{1 - exp(-6Z_*)}. \tag{2}$$

" (see P4L3-22)

*Then again on page 8, line 2, the authors analyze the results in terms of some chosen crystal size class that was dominant in some previous studies. Since the model calculated the size distribution (I think), the authors can interpret their results in terms of it. Picking a particular size class is odd because one of the sensitivity experiments involved changing the size classes (i.e. the discretization of the crystal size distribution). Note that the growth rate depends on crystal size, so a simple*

20  *average size might not be appropriate.*

*In the conclusions, page 10, line 2, the authors mention the 'complicated form of the relationship depending on suspension index'. I think this relationship might become clearer through thinking about changes to the crystal size distribution.*

*7. Section 4. I felt the explanations could have been clearer. For example, phrases like 'supercooling is utilised more efficiently' seem to be key but I didn't understand precisely what this meant or why it happened.*

25  *9. The figures contained a lot of information so they are necessarily somewhat complicated. However, all figures are particularly difficult to interpret printed in black and white. In figures 7–10, I would have coloured each point according to $Z_*a$, rather than dividing into bands. The complicated legends could then be replaced by a simpler colour bar.*

Thank you for these very important comments! We admit that naively picking an intermediate crystal size is unreasonable in

30  this study. However, it is unfortunate that we did not output the size distribution in the steady state before. Consequently, we have rerun the extensive sensitivity experiments to obtain the size distribution information, and have found results that are

interesting. We have adopted a new method to calculate $Z_*$ based on the size distribution information, rather than choosing a particular size class, and have described the method on P9L14-18:

"

[revised manuscript text omitted]

"__P9L18-P10L8

"

15    As expected, the complex response of $C_i$ to variations in $T_{SC}^0$ (Fig. 9) is similar to the relationship between $T_{SC}^0$ and thickening rate (Fig. 7) in the VM model.

"__P10L18-19

As the referee expected, the relationship become clearer through thinking about changes to the crystal size distribution. In addition, we redrew the original Fig. 10 in a more appropriate style:

20    "

The magnitude of the difference in $C_i$ calculated by VM and VU models (VM minus VU) is compared in Fig. 10a, where we find that $C_i$ calculated by the VM model is always larger than that calculated by the VU model. In general, the difference increases with decreasing $\bar{Z}_*$, while the sensitivity grows with increasing $D_{SC}^{ini}$. The dependence on $\bar{Z}_*$ is once again due to the impact of the combined thermodynamic processes, i.e., the efficient growth in the upper supercooled part of the plume

25    together with the maintenance of supercooling by melting of frazil in the lower part, discussed above. We also see similar behavior for the difference in the thickening rate (Fig. 10b).

"__P10L21-26

    5. Page 4, around line 15. There is a pair of papers (Rees Jones and Wells, 2015 & 2018, the latter of which you cite) which

30    update the treatment of frazil crystal growth/melting.

These interesting studies focus on the details of frazil ice growth in supercooled water, but include no discussion of frazil melting within overheated ambient water. Consequently, the treatment of the melting frazil ice, adopted in this study, remains the best available, and has been widely used in a series of previous studies.

Rees Jones and Wells (2015) carried out a simulation of microphysical processes during frazil growth in supercooled water, and their model result is consistent with the parameterization of frazil growth proposed by Jenkins and Bombosch (1995), which considered the aspect ratio of the frazil disc in the growth rate formulation. However, after Jenkins and Bombosch (1995), all subsequent studies did not encapsulate that factor into the frazil ice growth rate. Our formulation is consistent with those later studies.

In our study, the relationship between supercooling and thickening rate is strongly controlled by the suspension index that is determined by the drag coefficient (related to $u_*$) and frazil size (related to $w_i$). Our sensitivity experiments using a range of crystal sizes from "A" to "2×A" and different values of drag coefficient effectively characterise that relationship. Changing other parameters, such as $W_{ini}, C_i^{ini}, V_a, \bar{n}$, has little impact on the relationship, because the corresponding suspension indices changes little from that in the standard run. Therefore, we would argue that adopting an alternative growth rate formulation would not alter the form of the relationship, despite giving a faster frazil growth rate. That effect is alternatively captured by using larger frazil ice sizes, as discussed in Rees Jones and Wells (2018), so has been effectively examined in our 211 sensitivity runs.

In summary, we retained the commonly-used formulations for frazil ice growth/melting in order to focus on the impact of the introduction of the vertical profile of frazil ice concentration.

*Regarding the need to limit the mass loss due to melting, I think this is a departure from the 'equilibrium' (i.e. steady state) assumption in your vertical ice concentration distribution. Because the local melting rate is proportional to local concentration, the ice mass should just approach zero exponentially over time. Or perhaps there is an issue with your time stepping scheme or use of an excessively large time step (how was 25 seconds chosen)?*

The concept of an equilibrium vertical profile of frazil ice was introduced from suspended sediment dynamics (Cheng et al., JPO, 2017). However, the equilibrium vertical profile of suspended sediment can hardly be reached in realistic conditions, since the deposition process can make the vertical profile deviate from the equilibrium state (Cheng et al., ESPL, 2016). For suspended frazil ice, not only its melting in the lower overheated part of the ISW plume, but also its growth in the upper supercooled part, make this issue more complicated. Both contribute to the departure from the equilibrium state, in addition to the deposition process. For this study, introducing the equilibrium vertical profile of frazil ice can be regarded as the leading-order approximation to the non-equilibrium one.

Referring to the time step, it must be very small to reconcile the frazil melting at all grid points in our two-dimensional McMurdo Sound model, because the size distribution and vertical distribution of frazil concentration for each size differ at each grid point. Using a smaller time step can stabilize our models at a price of increased integration time caused by the

multiple-sized frazil ice module. We found that the use of 25 s, with the introduction of the limit on melting, was a good compromise that avoided model instability but also maintained the computational efficiency necessary for our extensive sensitivity experiments.

*6. Sensitivity experiments. In equation (3), there are a couple of 'fudge factors' (solid fraction, volume change) that are not varied. Changing these would be a direct, linear way to get more platelet ice in runs that had less than in observations. Can the authors explain what process the volume change parameter is supposed to represent?*

Both solid fraction and volume change are factors related to the complex processes that take place after the frazil ice precipitates onto the sea ice base. That issue is not our focus, and is not addressed in this study. Therefore, the adopted values for these uncertain factors follow Hughes et al. (2014) that is, to our knowledge, the only literature referring to the simulation of platelet layer thickness in McMurdo Sound based on the ISW plume concept. Setting the solid fraction to 0.25 is observationally supported by Gough et al. (2012), but the volume change factor is quite uncertain. As Hughes et al. (2017) argued, *"the subice platelet layer is composed of platelet crystals that, on average, doubled in volume after precipitating. This third factor is a broad estimate, as there appear to be no values from the literature to guide it."*. In view of that, we just kept these two little known factors, not directly related to the frazil-ice-laden ISW plume processes we are concerned with here, constant.

*More broadly, the authors seem to tune a large range of model parameters. This limits the predictive value of the study since these parameters are unknown a priori. Could these parameters be used elsewhere or would one need to retune each time? Is the greatly increased drag coefficient (for example) plausible?*

The large number of physical and input parameters that need to be tuned, with little observational guidance, is a common feature of models of the sub-ice environment, especially those focussed on frazil ice processes. However, reproducing the observational supercooling level and platelet layer thickness was not the only purpose of our study. More importantly, we aimed to investigate the relationship between supercooling and platelet layer thickening rate, and we expect those results to be more transferable to other geographic locations. We would expect that, if we apply this model elsewhere, such as underneath the ice shelves to simulate the marine ice accretion there, retuning will be necessary, but will be made easier by the knowledge acquired in this study. On the specific point of the drag coefficient, the increased values used here are motivated by the observational evidence from Robinson et al. (2017).

*Some of the sensitivity experiments seem very odd. Particularly changing the frazil size configuration (discretization of the crystal equations in size space). If the distribution were well resolved, changing the discretization wouldn't change the results. I think the author should use more crystals size classes (at least as a test) and should use a smaller minimum crystal*

*size. The minimum size is often very important in these models because it ends up being the size of nucleated crystals. A minimum size of 0.2 mm, therefore, seems large.*

Another important comment! "It is unlikely that the natural process of sediment transport by flowing water will be understood in precise dynamical terms in the foreseeable future.", said R. A. Bagnold (1980), a famous sedimentology scientist. However, as we know, the suspended frazil ice seems to be even more complicated than the suspended sediment because of the thermohaline exchanges, which has a number of uncertainties, including its size configuration. It could be set to an arbitrary configuration, even a single size for some idealized case studies (such as Jordan, J. R., Holland, P. R., Jenkins, A., Piggott, M. D., and Kimura, S.: Modeling ice-ocean interaction in ice-shelf crevasses, J. Geophys. Res.-Oceans, 119, 995-1008, http://doi.org/10.1002/2013JC009208, 2014 and Jordan, J. R., Kimura, S., Holland, P. R., Jenkins, A., and Piggott, M. D.: On the conditional frazil ice instability in seawater, J. Phys. Oceanogr., 45, 1121-1138, http://doi.org/10.1175/JPO-D-14-0159.1, 2015). This is because there is, to our knowledge, no direct observation about the frazil ice size configuration within the boundary layer underneath the ice shelves or the sea ice. However, following the referee's suggestion, we also conducted an additional 24 model runs with a wider size range, 0.05 to 3 mm divided into 8 classes, i.e., 0.05, 0.2, 0.7, 1.2, 1.7, 2.1, 2.6, and 3 mm, using three values (0.01, 0.02, and 0.03) for the drag coefficient.

[Figure]

Figure R: Relationship between $T_{SC}^0$ and thickening rate classified by outflow supercooled layer thickness $D_{SC}^{ini}$ for the wider size range runs. Red solid line depicts the upper limit of the data points shown in Figure 7 of the paper. Numbers in the legend represent the values of $D_{SC}^{ini}$. Triangle corresponds to the standard run.

It can be seen from Fig. R that the thickening rate for the wider size range is well beyond that of all the other runs. The lowest thickening rate for each $D_{SC}^{ini}$ corresponds to the lower limit of the drag coefficient $C_D$, because the low drag coefficient implies weak turbulence so ice crystals can easily be lost by the plume and few reach the lower part of the plume to melt and sustain the supercooling. However, for the wider size range runs, the lower limit of the drag coefficient is set to

5  0.01 that is also the lower limit of the observational estimates in McMurdo Sound, corresponding to a thin layer of tiny platelets with limited supercooling (Robinson et al., 2017). Although $D_{SC}^{ini} =78$ m and $C_D =0.01$ give a calculated $T_{SC}^0$ of 0.02 ºC for the wider size range, comparable with that for the standard run (0.023 ºC), the thickening rate is 2.7 times larger than that for the standard run, which reproduced the observational supercooling level (Fig. 4a) and platelet layer thickness in McMurdo Sound (Fig. 6a and b). Therefore, in view of that contradiction between the significantly larger platelet layer

10  production than suggested by observations, even using a drag coefficient appropriate for a thin, possibly transient platelet layer (Robinson et al., 2017), we infer that the wider size range gives a poorer representation of conditions in McMurdo Sound and we focus exclusively on the standard size range in our study. Nevertheless, we note that the form of the relationship between $T_{SC}^0$ and thickening rate shown in Fig. R is consistent with that shown in Fig. 7a when $D_{SC}^{ini} \geq 65$ m. Furthermore, the decreasing thickening rate transitions to an increasing trend with increasing supercooling for $D_{SC}^{ini} \geq 78$ m,

15  as for the standard size range runs (Fig. 7a).

Finally, we emphasize that our sensitivity experiments have included runs with changes in the average number of frazil crystals per unit volume, and the minimum crystal size in the standard size range runs is different from each other (Table 3), and both of those will affect the effect process secondary nucleation. However, we find that those parameters have comparatively little impact on our results.

*8. The use of acronyms was excessive for my taste. For example, page 5, line 24 contained five acronyms. I would recommend using more sparingly. Personally, I would change SIPL to 'platelet layer'; FIC to 'frazil concentration'; ASTR to 'thickening rate'. I would change VM and NVM (in any case NVM is an odd name, a double negative, perhaps 'vertically uniform' would be better). In a similar vein, the notation is excessively complicated, with an over-use of 'modifiers' (to give*

25  *a couple of examples, among many, TSC 0 , Z∗a).*

As suggested, we have changed SIPL to 'platelet layer'; FIC to 'frazil concentration'; ASTR to 'thickening rate', and used $T_{SC}^0$ and $\bar{Z}_*$ to represent the area-averaged supercooling level at the sea ice base and weighted suspension index, respectively.

30  *3 Technical comments*

*1. Title: I think the paper is broader than just the thickening rate*

Agree. We changed the title to "Responses of sub-ice platelet layer thickening rate and frazil ice concentration to variations in Ice Shelf Water supercooling in McMurdo Sound, Antarctica"

*2. Abstract: I would particularly avoid using so many acronyms in the abstract*

Agree. Following the specific comment above, the abstract contains only one acronym of "ISW".

*3. Abstract, line 16 'choice of frazil ice suspension index': is it really a choice? The meaning of the term 'suspension index' is not clear at this stage.*

Agree. We deleted 'choice of frazil ice suspension index' and gave a more explicit expression:

"…, and found to be predominantly controlled by the vertical distribution of frazil concentration." (P1L17-18)

*4. Page 1, line 25: maybe delete 'the' before 'elevated pressure'*

Revised

*5. Page 2, line 14: clause that starts 'in which supercooling' belongs in or immediately after previous sentence.*

Revised

*6. Page 2, line 24: how important are these differences (D not constant, 2 lateral dimensions) in the context of this paper?*

A variable ISW plume thickness (illustrated in Fig. 5a and b) allows the flow to be controlled by geostrophy. More importantly, we cannot obtain the two-dimensional patterns of hydrographic properties and platelet layer thickness using a one-dimensional model. Therefore, our two-dimensional model is much more appropriate for an investigation of the complex relationship between supercooling and thickening rate in McMurdo Sound.

*7. Page 3, line 5–13: split sentence*

Revised

*8. Page 3, line 6: 'can be quantified through' is vague. Perhaps 'is proportional to'*

Revised

*9. Page 3, line 8: Igr not clearly defined*

We revised the sentence as

"The frazil ice growth rate is found to be proportional to the following integral expression once a number of physical parameters within the commonly-used formulation of Jenkins and Bombosch (1995) are merged:"__P3L16-17

*10. Page 3, line 9: [0,1], use a comma*

Revised

*11. Page 3, line 27: don't understand Up + Ua or Up(Ua)? Similarly with V.*

The lower surface of the sea ice in McMurdo Sound is regarded as a horizontal plane. Therefore, Hughes et al. (2014) used a combination of the plume velocity and ambient current, $U_p + U_a$, to cope with the lack of a sloping ice base along which the buoyant plume ascends. This study follows that treatment.

*12. Page 3, line 28: italicize x and y (also elsewhere)*

Revised

*13. Page 5, line 7: reads slightly oddly because the 'outflow' is actually the inflow to your domain*

We revised "At the outflow the plume thickness …" as "The initial thickness of the ISW outflow from underneath McMurdo Ice Shelf …".

*14. Page 5, line 16: do you conserve latent heat in this the crystal volume doubling step?*

No. The crystal volume doubling occurs after the frazil ice precipitates out of the ISW plume onto the sea ice base. This process and its potential influences on the underlying ISW plume are not the considered in this study.

*15. Page 7, line 4–8: I suggest that you think about steady state solutions*

Our discussion of the relationship between supercooling and thickening rate focuses on the steady state attained at the end of the model runs.

*16. Page 8, line 18: presumably the consumption of supercooling is by the release of latent heat? If so, I would say this explicitly.*

Revised

*17. Page 9, line 13: 'the differences increase with decreasing Z∗a' seemed odd to me, because Za ∗ = 0 is supposed to recover the previous models*

As shown in Fig. 2b, VM returns to VU when and only when the suspension index equals zero and the whole ISW plume is vertically supercooled. That means VM never returns to VU in a practical application.

*18. Table 1: some quantities not defined in main text, including ar; n*

"$a_r$" and "$\bar{n}$" are two parameters associated with the frazil ice melting and secondary nucleation parameterizations and are not repeated in the main text (P5L22-27).

*19. Table 1: Tini seems to be a function of z, but I thought temperature was vertically averaged in the model?*

Sorry for the typographic mistake. We revised the expression of $T_{ini}$.as $-0.0573 \times S_{ini} + 0.0832 - 7.61 \times 10^{-4} D_{ini}$.

*20. Figure 3: annotations like p′ will be impossible to understand unless the reader is very familiar with the frazil literature. I think w′ is more usual than ω′ for frazil growth.*

The meanings of the annotations $f'$, $w'$, $N'$, and $p'$ are given in the main text, just before Fig. 3 is first referenced (P5L25-26). We changed $\omega'$ to $w'$ in Fig. 3 and the main text.

**Reply to Referee #2:**

*Summary*

*This manuscript identifies, and attempts to quantify, the complexity of the interactions between supercooled water, frazil ice*

5  *held in turbulent suspension, platelet ice accretion and the temporal evolution of Ice Shelf Water plumes originating in the*

*cold cavities of Antarctic ice shelves. The model used is an evolution of earlier models applied to this problem, with the*

*significant point of difference being inhomogeneous vertical profiles of supercooling and frazil-ice concentration.*

*In my opinion, the material covered could be of interest to the polar oceanographic community, and its presentation is timely,*

*considering the expansion of interest in the fate of the Antarctic ice shelves and the sea ice affected by their basal melt.*

10  *However, I do not believe that the stated 'main objective' has been satisfactorily achieved in light of the limitations of the*

*sensitivity studies. I also think that significant clarification is required before the manuscript would be ready for publication.*

We thank the referee #2 for his/her pertinent assessment of our paper and very constructive comments, which have helped us to improve our manuscript. We hope to address all the specific issues below.

*Major Points*

*1. Novelty of approach*

*a. The introduction of depth-dependent supercooling and frazil ice concentration was explored in a practical setting by Cheng et al., (2017), so it is not clear that there is a great deal of novelty in the present study.*

In Cheng et al. (2017), there is a simple application of VM underneath the western part of Ronne Ice Shelf to demonstrate its improved performance in reproducing the local observed marine ice thickening rate. However, there is no in-depth discussion of the impact of the combined nonlinear effects of the vertical profiles of supercooling and frazil ice concentration. To our knowledge, other than Hughes et al. (2014), there is no numerical modelling study of sub-ice platelet layer growth

25  using the ISW plume concept. We build on that study using a model with two horizontal dimensions, and more importantly, focus on the fundamental relationship between supercooling and platelet layer thickening rate in McMurdo Sound. We show for the first time how that relationship is controlled predominantly by the vertical distribution of frazil concentration.

*2. Confusion around terms and purpose*

30  *a. The general purpose of the study appears to be to improve model representation of marine ice accretion beneath ice shelves. That being the case, some justification is required for evaluating the model against an ISW plume beneath sea ice, which is generating a sub-ice platelet layer (as opposed to accreted marine ice). While there is clearly a lot of connection and similarity between these, some explanation of the differences and limitations of applicability need to be provided. For example: the difference in absolute pressure and its consequence for supercooling potential; the effect of basal slope for*

*generating buoyancy-derived momentum; the likely strength of flow and turbulence in the boundary layer and implications for size of crystal held in suspension.*

*b. A comment is also required on the general applicability of a model evaluated against a sub-sea ice ISW plume for the ice shelf cavity.*

5 *c. No definitions of the terms 'frazil ice' and 'platelet ice' are offered, and direct similarity between sub-ice platelet layers and accreted marine ice is implied. These terms are not interchangeable, and their use here contributes to a general confusion as to the overall purpose of the study. Use of the term ''platelet-like frazil crystals'' (p1 line 28) exemplifies this confusion (for both the authors and readers). Clear definitions of these terms would help, as would a statement that the plume in McMurdo Sound is being used to evaluate model performance on the basis that it is the most comprehensively*

10 *observed plume available, despite its expected differences (which should be clearly identified) from a sub-ice shelf ISW plume.*

*e. I find the comment in the Conclusions (p. 10, lines 9 – 11) that "additional observations of supercooled ISW, SIPL and frazil size spectrum in western McMurdo Sound are needed to constrain the relationships determining marine ice formation beneath ice shelves", rather strange. The recommendation surely should be to observe these parameters beneath ice shelves,*

15 *if that is indeed the desired outcome?*

We apologize for the confusions between marine ice and platelet layer under the ice shelves and sea ice, respectively, and between frazil ice and platelet ice. Conventionally, we refer to suspended ice crystals within the ISW plume as 'frazil ice', and refer to ice accretion underneath ice shelves and sea ice as 'marine ice' and 'platelet layer', respectively, throughout this

20 manuscript. In addition, we changed "platelet-like frazil ice crystals" to "disk-shaped frazil ice crystals". The general purpose of this study has been given in the previous response, and we only focus on the McMurdo Sound, where there exists a comprehensive observational database for model validation.

We do not entirely agree with the referee's statements about the distinction between the processes operating beneath ice shelves and platelet ice layers in front of ice shelves. Beneath the platelet ice layers the supercooling is produced by the

25 pressure drop experience by ISW at it emerges from beneath an ice shelf and rise towards the sea surface, while regions of marine ice accretion beneath ice shelves have typically very low basal slopes. We suggest that the main distinction is likely to be the magnitude of the vertical temperature gradient through the accreted ice. That will primarily affect only the post-depositional processes that we do not consider. The one indirect effect on the ISW plume might be an increased drag coefficient beneath the platelet layer, where more rapid freezing of the deposited crystals may create more irregularity in the

30 form of the ice-ocean interface.

To clarify these points, we revised the last section as follows:

"

In this study, we demonstrated how the vertical distributions of supercooling and frazil ice concentration within an ISW plume jointly determine the growth of suspended frazil ice, and thus the rate of platelet layer formation under sea ice and

marine ice beneath ice shelves. A vertically-modified, frazil-ice-laden, ISW plume model which encapsulates these combined nonlinear effects was applied to the McMurdo Sound region, and reproduced the observed ISW supercooling and platelet layer distributions in two horizontal dimensions. Using multiple model runs, the relationship of ISW supercooling to platelet layer thickening rate and frazil concentration in McMurdo Sound was explored, and shown to be dependent on the

5    suspension index that controls the vertical distribution of frazil concentration within the ISW plume. Moreover, when the thickness of a supercooled layer of ISW is large enough, the efficiency of converting ISW supercooling into frazil concentration, and thus platelet layer growth is determined by the suspension index. These findings highlight the need for further observations in McMurdo Sound, particularly focused near the ISW outflow region in the western sound, where the supercooled ISW plume and platelet layer are prominent, and more general observations that help to constrain the frazil size

10    spectrum within the sea ice-ocean boundary layer. In addition, the performance of the VM model in providing reliable estimates of supercooling and frazil ice flux at the platelet layer base makes it an attractive tool for coupling with sea ice models focusing on microscale processes within the bottom layer of the ice (Buffo et al., 2018).

It would be straightforward for the next step to investigate the relationship between supercooling and marine ice thickening

15    rate underneath ice shelves using the VM model. Quantifying this relationship would be the key to parameterizing the process in more complex three-dimensional, primitive equation ocean models, which frequently neglect details of the ice shelf-ocean boundary layer and processes associated with an evolving suspension of frazil ice crystals (Liu et al., 2017; Mueller et al., 2012, 2018). The main difference between the process of marine ice accretion beneath ice shelves and the growth of the sub-ice platelet layer discussed here is likely to be the magnitude of the vertical heat flux that the deposited

20    frazil ice is subjected to. At the base of an ice shelf, typically several hundred meters thick, the vertical temperature gradient is comparatively small, so the deposited crystals form a slushy layer (Engelhardt and Determann, 1987) that slowly consolidates, possibly as much through compaction as freezing. The ice-ocean interface and the associated drag coefficient are therefore likely to be very different to those observed in McMurdo Sound, where the platelet layer appears to comprise a more open matrix of ice and water that consolidates by freezing as heat is lost to the atmosphere. Therefore, the VM model

25    would need to be re-evaluated against observations of sub-ice shelf ISW plumes and the ice shelf-ocean boundary layer. Finally, further process studies, including the influence of the vertical current structure within either the ice shelf or sea ice - ocean boundary layer (Jenkins, 2016; Robinson et al., 2017) could also contribute to improving our understanding of marine ice and platelet layer formation.

”__P11L2-31

30    Moreover, as suggested we also added a statement that

“…, arguably one of the most comprehensively observed ISW plumes available (HU14; Langhorne et al., 2015; Robinson et al. 2014).”__P2L26-27

*d. The general confusion of purpose is demonstrated in that the Abstract opens with the ISW plume in McMurdo Sound (I would argue that the abstract should focus on the model rather than its application), while the Introduction is primarily concerned with the sub-ice shelf regime.*

5 As suggested earlier, the VM model used here cannot be claimed to be novel in this study. Therefore, we would argue that the abstract should focus on our findings revealed by applying VM to McMurdo Sound rather than the model itself. Furthermore, a number of papers (Hughes et al., 2014; Langhorne et al., 2015; Robinson et al., 2014, 2017) focused on issues closely related to this study, have discussed ISW plume processes under the ice shelves in their Introductions before the authors focus McMurdo Sound as an analogue. We have done likewise, in order to inform readers unfamiliar with the

10 related issues about the potential wider applicability of the study.

*3. Inadequate explanation of sensitivity studies and choice of values for parameters*
*a. Page 5, Line 26-27 justifies the tuning of parameters to reproduce the observed SIPL and supercooled ISW thicknesses, but we are not given an indication of the strength of this tuning. How do the chosen parameters for the standard run*

15 *compare with the observations, and observed range of values?*

The large number of physical and input parameters that need to be tuned, with little observational guidance, is a common feature of models of the sub-ice environment, especially those focussed on frazil ice processes. Therefore, the final adopted values for the parameters in the standard run must be determined by trial and error, guided by accumulated experience with

20 the model. We modified the text to "…, extensive tuning of the least constrained model parameters, including…" P7L9.

*b. I would expect that the sensitivity of the model could be just as significant for ar, Nice and solid volume fraction as for the parameters that were included in the sensitivity study. Further testing of the sensitivity to these parameters would be ideal, but at least a statement acknowledging (and justifying) their exclusion is required.*

The aspect ratio of frazil discs $a_r$ is usually fixed at 0.02, based loosely on laboratory observations, in frazil-ice-laden ISW plume modeling (Holland and Feltham, 2005, 2006; Hughes et al., 2014; Smedsrud and Jenkins, 2004), and is only associated with frazil ice melting in this study. The relationship between supercooling and thickening rate is strongly controlled by the suspension index that is determined by the drag coefficient (related to $u_*$) and frazil size (related to $w_i$).

30 Our sensitivity experiments using a range of crystal sizes from "A" to "2×A" and different values of the drag coefficient effectively characterise that relationship. Changing other parameters, such as $W_{ini}, C_i^{ini}, V_a, \bar{n}$, has little impact on the relationship, because the corresponding suspension indices change little from that in the standard run. Therefore, since changing the aspect ratio would hardly change the suspension index, we would expect little impact on the key relationships. The number of frazil ice sizes $N_{ice}$, is associated with many uncertainties, including the size configuration. Even a single size

has been used for some idealized case studies (such as Jordan, J. R., Holland, P. R., Jenkins, A., Piggott, M. D., and Kimura, S.: Modeling ice-ocean interaction in ice-shelf crevasses, J. Geophys. Res.-Oceans, 119, 995-1008, http://doi.org/10.1002/2013JC009208, 2014 and Jordan, J. R., Kimura, S., Holland, P. R., Jenkins, A., and Piggott, M. D.: On the conditional frazil ice instability in seawater, J. Phys. Oceanogr., 45, 1121-1138, http://doi.org/10.1175/JPO-D-14-

5    0159.1, 2015). This is because there are, to our knowledge, no direct observations of the frazil ice sizes within the boundary layer underneath ice shelves the sea ice. Nevertheless, as a test suggested by Referee #1, we also conducted an additional 24 model runs with a wider size range, 0.05 to 3 mm divided into 8 classes, i.e., 0.05, 0.2, 0.7, 1.2, 1.7, 2.1, 2.6, and 3 mm. The corresponding detail found in P14L32-P16L19 of this report. The solid volume fraction is related to the complex processes that take place after the frazil ice precipitates onto the sea ice base and that are not addressed in this study. It is just a

10   proportionality factor in the calculation of the thickening rate, and does not exert any influence on the frazil-ice-laden ISW plume processes that we are concerned with here.

*c. The role of settling dynamics is clearly significant here (as in sediment studies), but this is not referred to at all.*

*Minor Points*

15   *The thickness of the accreted platelet layer is a critical parameter for testing the performance of the model. However, the mechanism employed for transferring crystals out of the suspended frazil and into the accreted platelet ice layer is not made explicit.*

We improved the description of the frazil ice precipitation as

20   "

We treat the frazil ice precipitation rate $p'$ as inverted sedimentation and follow the parameterization of McCave and Swift (1976):

$$p' = w_i C_i \left(1 - \frac{U^2}{U_c^2}\right) \times He \left(1 - \frac{U^2}{U_c^2}\right), \tag{3}$$

where $U_c$ is a critical velocity, above which precipitation cannot occur, determined by Jenkins and Bombosch (1995):

25   $$U_c^2 = \frac{\theta_i(\rho_0 - \rho_i)g2r_e}{\rho_0 C_D},$$

where $\theta_i$ is the Shields criterion, $\rho_0$ and $\rho_i$ are reference seawater and ice densities, respectively, $g$ is gravity, $r_e$ is the equivalent radius of a sphere with the same volume as the frazil disk. The frazil ice rise velocity, $w_i$, is calculated by Morse and Richard (2009):

$$w_i = \begin{cases} 2.025 D_i^{1.621} & if \ D_i \leq 1.27 \ mm \\ -0.103 D_i^2 + 4.069 D_i - 2.024 & if \ 1.27 < D_i \leq 7 \ mm \end{cases},$$

30   where $D_i = 2r_i$ is the diameter of a frazil crystal in mm. The inclusion of the Heaviside function $He$ means that negative precipitation (i.e., erosion of previously deposited frazil ice) is not permitted. Because we have no idea about how cohesive the ice crystals are once they have settled, the estimation of an erosion rate would entail additional uncertainties.

The complex processes after the frazil ice precipitates onto the sea ice base are simplified in our model. In order to calculate platelet layer thickness $D_P$ at the $n^{th}$ time interval, we adopt the assumptions of HU14 that solid ice fraction within the platelet layer in McMurdo Sound is 0.25 based on the observational estimation from Gough et al. (2012) and that the ice crystals, on average, double in volume after precipitation:

$$D_P = \frac{1}{0.25} \times 2 \times \sum_{k=1}^{n}(p'_k \times \triangle t).$$

It should be noted that the volume change factor is a broad estimate, with almost no supporting evidence in the literature to guide it.

"__P6L14-P7L2

*The manuscript is rather heavy-handed in its use of acronyms:*

*'ISW' is well-known and can be used freely;*

*The reader may need a few reminders of what 'SIPL' represents;*

As Referee #1 also suggested, we changed 'SIPL' to 'platelet layer'.

*Please expand 'MMS' to McMurdo Sound in all cases;*

Revised

*'FIC' should not be used, especially as it is variously interchanged with 'ci';*

As Referee #1 also suggested, we changed 'FIC' to 'frazil concentration'.

*'VM' is OK;*

*but 'NVM' is non-intuitive;*

As Referee #1 also suggested, we changed 'NVM' to 'VU' (vertically-uniform).

*'ASTR' is non-intuitive – is it possible to avoid this altogether?*

As Referee #1 also suggested, we changed 'ASTR' to 'thickening rate'.

*The superscript '0' (as in T0SC) is not explained and left to the reader to figure out.*

Revised ("…. at the sea ice base ($T_{SC}^0$, superscript "0" denotes the sea ice base)…."__P7L22).

*P 1, Line 27: "a necessary condition for ice crystals to form…" either needs a reference or should be removed, as primary nucleation is not thought possible under observed supercooling conditions.*

We deleted the words "form and" from the sentence.

*P 2, Line 5-6: "the SIPL should not be ignored…". This manuscript is not offering new evidence on the role of the SIPL, so is not the place to be offering this recommendation.*

5    We revised "In addition, the presence of the platelet layer can also exert an influence on the estimation of sea ice thickness from freeboard information obtained by satellite altimetry (Price et al., 2014). Thus, the platelet layer should not be ignored when investigating sea ice thickness near an ice shelf front using either numerical models or remote sensing of surface elevation." to "Therefore, the platelet layer should not be ignored when investigating sea ice thickness near an ice shelf front." (P2L5-6).

*P2, Line 8-9: "Owing to the paucity of direct observations…" And yet this manuscript relies heavily on observations collected in McMurdo Sound. I do not feel that the observations available (beyond HU14) are sufficiently acknowledged or applied.*

15    We revised "The observations, including both oceanographic and ice core data (Fig. 1), are taken from HU14." to "To our knowledge, the data reported by HU14 are the most comprehensive available to evaluate our model, including both oceanographic and ice core data in two horizontal dimensions adjacent to McMurdo Ice Shelf." (P7L6-8)

*P2, Line 16-17: There is no marine ice production in McMurdo Sound (this is an example of the general confusion identified*
20 *above).*

We revised the wording to "Consequently, earlier assessments of either marine ice or platelet layer production in the aforementioned areas may need to be reevaluated." (P2L23-24)

*P2, Line 25-26: here it is inferred that the ISW plume thickness is allowed to evolve in the present study, but this is not stated*
25 *explicitly.*

We made this explicit by adding an adjective as "The unsteady VM and VU models used in this study are …". (P5L18)

*P3, Line 5: The statement that "the concentration of suspended frazil ice controls the dynamic and thermodynamic evolution of ISW outflows" requires a reference.*
30    Added (P3L15)

*I see that a crystal size distribution, incorporating 5 size classes, is being used. But how this is incorporated is not explicitly stated.*

*a. P 5, Line 13: I am not convinced that the frazil ice crystals should be evenly distributed among size classes. Naively, I would expect a much greater proportion to reside in the smallest size class(es). At the very least, this treatment needs a reference.*

5   The even distribution be cited in Holland and Feltham (2005, 2006), Smedsrud and Jenkins (2004) (added in P6L12). In fact, this treatment is only applied to the ISW outflow boundary, after which the overall size distribution is freely evolving.

*b. Is the smallest size class small enough? (I'm not sure on this). Is there a reference to support this?*

*d. I think more justification is needed on the collapsing of a range of size classes to a single size class for comparative*

10  *purposes.*

*c. I cannot see anywhere that explicitly states whether crystals move between size classes as they grow/melt; or if and how they are removed from the largest size class to accrete as part of the SIPL.*

Again, the frazil size configuration, including the number of frazil ice sizes $N_{ice}$ and minimum size, must be set arbitrarily

15  because of the lack of observations to guide the choices. During the model validation, we have tried many size configurations with a range of settings for the minimum size. However, we found it hard to obtain a good match between the model and observational data using a smaller minimum size. In response to Referee #1's comments we no longer analyse the results in terms of a single size class, but instead characterise the size distribution through the average suspension index. We apologise for not explicitly stating that the frazil crystals grow by transferring between size classes. In fact, in this study, we

20  adopted a commonly used scheme proposed by Smedsrud and Jenkins (2004) to calculate the transfer processes induced by freezing and melting between different size classes. To make this fact clearer, we added a statement "The frazil ice size distribution is represented by 5 crystal size classes, and the transfer processes, induced by frazil freezing and melting, between different size classes are calculated using the scheme proposed by Smedsrud and Jenkins (2004)." to the model description in P6L9-11.

*Which temperature is being used? (i.e. in-situ, potential, conservative: : :) In observations of the ISW plume in question (e.g. Dempsey et al., 2010; Robinson et al, 2014; which the authors make reference to), the upper ocean is homogeneous in potential temperature.*

Potential temperature. The data sets taken from Hughes et al. (2014) in this study also give potential temperature. We have

30  made that explicit in P5L21.

*I presume that latent heat is being added as ice crystals grow, since the supercooling is variously referred to as being 'released', 'utilized', 'converted' and 'varied'. However, how this is incorporated is not identified or referred to anywhere.*

The governing equations and a variety of parameterizations including the frazil ice growing/melting formulations are not given in the main text, since they are presented in detail in the cited literature, and our implementation follows those descriptions. The following text cites the relevant papers:

"

5 Both VM and VU models combine the same commonly-used parameterizations of thermohaline exchanges across the ice–water interfaces, specifically a three-equation formulation (Holland and Jenkins, 1999) for the sea ice base and a two-equation formulation for frazil ice (Galton-Fenzi et al., 2012), with a multiple size–class frazil dynamics model (Smedsrud and Jenkins, 2004), to calculate basal freezing ($f'$) and frazil melting/freezing ($w'$), secondary nucleation ($N'$), and precipitation ($p'$). These processes are summarized in Fig. 3. Rather than repeat all the equations here, we recall some of

10 them and present how we set up our ISW plume models on the McMurdo Sound domain.

"__P5L22-27.

Moreover, this comment is similar with the one proposed by Referee #1:

*16. Page 8, line 18: presumably the consumption of supercooling is by the release of latent heat? If so, I would say this explicitly.*

15 We revised the text as ", where melting of the crystals can mitigate the release of latent heat (Fig. 2b)." (P9L25)

*P3, Line 27: To aid clarification, could the word 'background' be changed for 'ambient' (current), since this will align with the subscript used in the symbol.*

Revised

*P3, Line 28: I take from this that the 'tidal' speed is not varying in time, but is effectively just an additional source of kinetic energy?*

Yes. This is the conventional treatment of tidal effects in the literature so far.

25 *P 4, Line 8: Please identify where the use of DSC = 50 m is derived from. Is there observational support for this? Or has it come from HU14?*

Setting $D_{SC}$ to 50 m is arbitrary, because the aim here is only to give a schematic comparison between the calculated $I_{gr}$ (i.e., frazil ice growth rate) by the VM and VU formulations shown in Fig. 2b, but is still within the calculated range for the

30 standard run (Figs. 1, 5c and d). We revised the sentence as "…, where $D_{SC}$ = 50 m (a value within the calculated range for the standard run, Fig. 1) in all the cases." (P5L2)

*P 4, Line 13-14: I am confused as to why you would want to avoid frazil melting in the lower part of the plume if it is overheated.*

Sorry for this puzzling expression. We revised the text "for given supercooling, in order to avoid frazil melting in the lower, overheated part of the ISW plume, $Z_*$ must be large enough to maintain greater frazil concentration in the upper, supercooled part." as "…, if $Z_*$ becomes larger, there is higher (lower) frazil concentration in the upper (lower), supercooled (overheated) part of the ISW plume." (P5L8-9)

*P 5, Eqn (3): the fact that individual ice crystals may double in volume after precipitation shouldn't affect the thickness of the SIPL, since presumably their growth is into the interstitial spaces, and shouldn't cause the entire SIPL to expand?*

The volume change factor related to the complex processes after the frazil ice precipitates onto the sea ice base is not the issue we focus on, and cannot be addressed in this study. Therefore, the adopted value for this uncertain factor simply follows Hughes et al. (2014) that is, to our knowledge, the only study of platelet layer thickness in McMurdo Sound based on the ISW plume concept. How the growth of deposited individual ice crystals affects the sub-ice platelet layer is not clear. As Hughes et al. (2014) argued, *"the subice platelet layer is composed of platelet crystals that, on average, doubled in volume after precipitating. This third factor is a broad estimate, as there appear to be no values from the literature to guide it.".*

*P5, Line 8: My understanding is that the 'background circulation' applied in HU14 was to provide the buoyancy-driven momentum that the sub-IS plume would naturally possess, but which cannot arise automatically within a model beneath a horizontal ice base (i.e. sea ice). That being the case, this represents a departure from sub-Ice Shelf applications, and should be explained again here.*

Buoyancy-driven flow beneath a horizontal ice base can be simulated if terms involving the gradient in plume thickness and/or density are introduced into the momentum balance. HU14 did not do that, so the "background circulation" was introduced to provide a flow. We added further clarification of that point to the text: "…: the former, which represented the only source of momentum in the study of HU14, is…" (P6L5)

*P 6, Line 10-11: I am not satisfied that there is sufficient support for the statement that 'both models reproduce the observed values of ISW supercooling reasonably well'. Looking at Figure 4a,*

*a. The structural difference between sites FN and NN is not captured at all;*

*b. The neap-tide supercooling at site C is very different to the modelled result, and the difference between spring and neap at that site is not reflected in the model timeseries;*

*c. The observed values at Site W do not coincide with the model timeseries values;*

*d. The only site that has 'reasonable' agreement is Site I, which is very close to the inflow point of the prescribed ISW plume.*

We admit all of the model discrepancies mentioned above, and would argue that they arise from the limitations in our model setup that we now explicitly refer to in the text:

"

It can be seen that at the end of the simulations both VM and VU models reproduce the observed reduction in ISW supercooling at the sea ice base ($T_{SC}^0$, superscript "0" denotes the sea ice base) in the cross- and long-sound directions, in spite of some evident model discrepancies (Fig. 4a) that may result from the limitations in our model setup: both the ambient current and tides are treated as temporally and spatially constant; there are no long-term observations of ISW outflow to provide reliable boundary conditions; we use a constant drag coefficient, ignoring the spatiotemporal evolution of the sea ice basal form characterized by the platelet layer. Nevertheless, the SS of $T_{SC}^0$ calculated using VM and VU models are 0.56 and 0.58, respectively, and the CC and RMSE are also reasonable (Table 2).

"__P7L21-27

*Also, the spatial resolution in the figures, especially of the ISW plume structure and SIPL thickness (figures 5 and 6) seem to greatly exceed the stated resolution of the model (1 km, P 5, Line 12). This then has implications for the statements made about how these are resolved by the model (as in P 6, Line 29). Also, this seems like fairly low resolution for the ISW plume itself, at 3 km wide.*

The colour map shown in Figs. 5 and 6 is interpolated from our model results using a Natural Neighbour method. We added this statement to the captions of Figs. 5 and 6. In addition, we still believe that "Such small-scale features in the platelet layer thickness distribution, if present, were not resolved by the relatively coarse spatial distribution of ice-core sampling (dots in Fig. 6). Nevertheless, the platelet layer thickness calculated by the VM model at drill sites agrees well with the measurements (Fig. 6a), …" (P8L13-15), because the resolution of our models, i.e., 1×1 km, is significantly higher than that of the ice-core sampling. The figure of '3 km wide' refers to the initial width of the ISW plume at the outflow. Downstream, the ISW plume extends all along the 'a-b' boundary in the steady state (Fig. 5). Therefore, to avoid this misunderstanding, we revised the descriptions of $D_{ini}\left(D_{SC}^{ini}\right)$ and $W_{ini}$ in Table 1 as "Constant ISW plume outflow thickness (constant outflow supercooled layer thickness)" and "ISW plume outflow width with constant $D_{ini}\left(D_{SC}^{ini}\right)$", respectively.

*P 7, Line 27ff: Behavior around the 'inflexion' points is particularly interesting. I think it would greatly strengthen the paper if the inflexion points could be investigated in higher resolution (especially in Z\* and σSC) to the point where the mechanism causing them could be adequately explained. There is an informal explanation offered in lines 28 – 30, but studying this in greater detail could be very useful for determining if this is a model artefact or a real phenomenon that could invite observational investigation.*

*P 9 Line 8: The sentence is incomplete? Perhaps ": : : there are analogous inflexion points: : :"?*

Thank you for these suggestions. Following Referee #1's related comments, we modified the paper significantly to reinforce our analysis on the inflexions. We adopted a new method to calculate $Z_*$ which involves the size distribution information, rather than choosing a particular size class, enabling us to analyze our results in terms of this new representative suspension index. We then redrew the corresponding figures with each point colour-coded by the new representative suspension index rather than dividing into bands, and found more details that are interesting and beneficial for our discussion. See our earlier response to Referee #1's related comment (P?L? of this response):

*P 8, lines 9-10: perhaps the wording could be more straightforward? Suggest using "... because the supercooling is used less efficiently for producing SIPL in the NVM than in the corresponding VM runs."*
Revised

*P 8/9 How do the critical thicknesses of the supercooled portion of the ISW plume (65 and 78 m) compare with observed thicknesses of the McMurdo Sound ISW plume?*

There are no direct observations of the ISW outflow from underneath McMurdo Ice Shelf. Nevertheless, as we have mentioned in the main text (P9L7-8), "… $D_{SC}^{ini}$ =65 m is the value estimated by HU14 based on the measurements conducted by Lewis and Perkin (1985) and Jones and Hill (2001)."

*P 9 Line 24: The sentence is incomplete? ": : : that have been taken instruments not specifically designed: : :"*
Revised

*P10 Line 14-15: Robinson et al. (2017) is an example of such a process study and could be referenced here.*
Added

*Table 1: Is there support in the literature for choosing ar = 0.02?*
Yes. We have mentioned in the response to comment 3-b: "The aspect ratio of frazil discs $a_r$ is usually fixed at 0.02, based loosely on laboratory observations, in frazil-ice-laden ISW plume modeling (Holland and Feltham, 2005, 2006; Hughes et al., 2014; Smedsrud and Jenkins, 2004), …"

*Table 3: Could the names of the variables be repeated here so that the table is self contained?*
Revised

*Figure 1: Why does the model data only extend as far as the borehole domain, and not to the edges of the model domain?*

Revised as suggested.

*Figure 1: Do the location names (C, I, W, NN, FN) mean anything in the present study? If yes, can these be provided? If no, could they be simplified to A -> E without loss of generality?*

5   The location names follow Hughes et al. (2014), and we added their definition to the caption of Fig. 1

*Figure 1 caption: The 'oceanographic and SIPL data' are not shown in the figure, so reference to them should be removed from the caption. Unless what is meant are the locations of the oceanographic and SIPL data?*

Removed

*Figure 2b: The almost-vertical blue line to the right, and horizontal blue line along the x-axis appear to be mistakes?*

Sorry, we do not find these mistakes in Fig. 2b. Maybe a different edition of PDF software?

*Figure 2b: It would be helpful to add the words 'Melt' and 'Freeze' alongside the appropriate parts of the y-axis. Also, I*
15   *think that the change of vertical scale is not necessary (and possibly only adds confusion)*

We added the necessary words. It should be noted that the negative (melting) integral value is an order of magnitude larger than the positive (freezing) one, so if the vertical scale is unified, the positive part becomes very compressed.

*Figure 3 caption: "::relevant processes within a supercooled ISW plume of homogeneous temperature and salinity".*
20   Revised

*Figure 6: Just a suggestion: it would aid comparison if the location dots could be colored by the measured SIPL thickness.*

Revised

25   *Figure 7a: There is no explicit specification of what 'VM30' through 'VM110' refers to.*

Added to the caption of Fig. 7

*Figure 7a and 7b: Text says that the ASTR estimate is from the last 30 days of the model runs. This needs to be repeated here.*

Added

*Also, converting the ASTR to cm/day may be a more useful unit (and less misleading, given that it only refers to 30 days' worth of growth)*

Revised

*Figure 9: The formulae for the fitted lines could be moved out to the right so as not to become lost amongst the data points.*

Fig. 9 has been modified (see earlier response to Referee #1).

*Figure 10: I don't think the additional enlargements provide any further useful information and could be removed.*

5    *Figure 10: Numbers on axes are too small.*

Fig. 10 and the accompanying text have been modified (see earlier response to Referee #1).

**Reply to Ken Hughes**

*I (Ken Hughes) was lead author of the paper on which a number of comparisons are made in the current manuscript. Consequently, I have a unique perspective on this paper. I have included a number of comments that would improve the*

5 *clarity of the manuscript, which I read with interest. I am not, however, an official reviewer, and so am providing these comments for the authors' reference.*

*The authors modify an existing two-dimensional, time integrated ISW plume model to (i) test its ability to recreate observed supercooling and SIPL thickness observations and (ii) via a rigorous sensitivity study, investigate which parameters have disproportionate influence on the spatial distributions.*

10 *The idea behind the study is worthwhile as McMurdo Sound provides arguably the best location against which an ISW plume model can be test. Indeed, the two-dimensional model is a noticeable improvement over the 1D approach used by Hughes et al. (2014).*

First, we thank Ken for his paper (Hughes et al., 2014) greatly helping us complete this study. We also thank Ken for his

15 pertinent assessment of our paper and very constructive comments, which have helped us to improve our manuscript. We hope to address all the specific issues below.

*Major issues*

*Line 5.27 "some tuning of model parameters" overlooks some key details*

20 *1. Why is the outflow only 3km? Observations in Hughes et al have ISW outflow adjacent to the ice shelf across the whole Sound.*

We experimented with a range of different outflow widths, and found that 3 km gave the best match to observations of platelet layer thickness. The plume rapidly spread across the width of the domain (Figure 5), so the results are consistent

25 with the observations of HU14.

*2. There is no mention that frazil ice classes in this paper are well below the mode size predicted by Hughes et al. (2014) of 2.5mm radius*

30 It is hard to compare directly the frazil ice classes of this study with those of HU14 in view of the many differences both in the treatment of the plume dynamics and the frazil ice processes. Since the crystal size required to reproduce platelet layer thickness is so strongly dependent on these other aspects of the model, and there are no observations of the dimensions of suspended crystals, we chose not to emphasize these differences in model setup.

*Line 2.25 Somewhere in the paper, a description of the plume dynamics beneath the sea ice is warranted. Hughes et al. (2014) used a constant plume thickness out of necessity, because one-dimensional models can't easily deal with a lack of forcing (ie a gravitational component beneath a slope). It appears from Figure 1 that the plume in this paper tends toward geostrophic balance. Is this true? Put another way, does having two spatial dimensions and the time dimension avoid the issues faced by Hughes et al.*

Adding terms to the momentum balance that include the horizontal gradients of plume thickness and/or density avoids the issue. That cannot be done in one-dimensional models, although it adds complexity. Adding the second horizontal dimension allows the spatial distribution of the platelet layer to be simulated. We added a statement that "Finally, it can be seen that in both models the ISW plume flow is predominantly governed by a geostrophic balance (Fig 5a-d)." (see P8L1-2)

*Line 6.29 Although the overall spatial distribution of SIPL may be graded as "excellent", it predicts a large areas where the thickness is 15m, twice the observations. Similarly, the "poorly" graded NVM model is closer to observations near the outflow. In other words, comparison between the models in terms of SIPL is not a simple as calling one "excellent" and the other "poor".*

As we have stated in the main text, the coarse resolution of the ice-core sampling cannot resolve the simulated high gradient of platelet layer thickness near the ISW outflow. It remains unknown whether such sharp gradients exist. Where observations exist, the VM model has superior performance in reproducing the platelet layer thickness, compared with the performance of VU model.

*There is no citation, let alone discussion, of Holland and Feltham (2005, doi:10.1017/S002211200400285X), which specifically deals with the issue of the vertical distribution of frazil ice.*

This point is consistent with one of the Referee #1's comments. We have now cited this paper in P2L21-23: "Idealized one dimensional models confirm that the vertical distribution of frazil concentration cannot remain well-mixed in the upper layers of the ocean (Svensson and Omstedt, 1998) and beneath ice shelves (Holland and Feltham, 2005)."

*On a similar note, there appears to be no real attempt to constrain Z\*, the suspension index. A large range is proposed, but the reader is not given any idea what a reasonable value is.*

The major purpose of this study is to investigate the relationship between the supercooling and platelet layer thickening rate in McMurdo Sound. This complicated relationship is strongly controlled by the frazil ice suspension index. To this end, we

highlight the need to improve frazil ice observations within the sea ice-ocean boundary layer in McMurdo Sound, in order to provide some constraint on the suspension index.

*Minor issues*

5  *Line 1.13: this is not a new frazil-ice-laden ISW plume model, but rather an existing ISW plume model with an improved frazil-ice treatment.*

We changed "new" to "vertically-modified".

*Line 1.15: This sentence should written as the dependence of supercooling in MMS on the SIPL thickening rate, not the other*
10  *way as it is currently. In other words, the current sentence says the dependence of x on y is investigated, yet two sentences later, it says x can be expressed as a function of y.*

*Alternatively, reword to something like "Using this model, we test how ISW supercooling in MMS influences SIPL thickening rate".*

*This issue also arises for the heading of Section 4.2*

We intended to discuss the dependence of platelet layer thickening rate and depth-averaged frazil ice concentration on ISW supercooling in McMurdo Sound. The sentence "For each suspension index, SIPL thickening rate can be expressed as an exponential function of ISW supercooling.", has been removed, as a result of other changes in the main text (see our response to the comments of Referees #1 and #2 in detail).

*Line 2.10 A reference to Galton-Fenzi et al. (2012) is warranted in this paragraph as they develop a three-dimensional model with frazil ice dynamics. It is not a plume model, per se, but it is does involve a non-vertically-uniform approach to frazil ice dynamics.*

Added in P2L10-11.

*Line 2.16 There is a slight issue with using MMS as an example for areas where marine ice production should be reassessed as it is no longer termed marine ice beneath sea ice.*

This comment is similar to one of the major concerns of Referee #2. We are sorry for the blended usage of these specific
30  terms in this study. The revised paper now, consistently refers to suspended ice crystals within the ISW plume as 'frazil ice', and to ice accretion underneath ice shelves and sea ice as 'marine ice' and 'platelet layer', respectively.

*Line 3.16 Hughes et al. (2014) improved upon the method of using T_sc at mid-depth. This is noted by Cheng et al. (2017) but is not noted here.*

We added the following statement "It is worth mentioning that in order to take the supercooling into account when $\sigma_{SC}<0.5$, HU14 integrated $T_{SC}$ over the supercooled part only without introducing any frazil ice melting. However, in this study, we will demonstrate that the important role of frazil ice melting in the lower, overheated part of the plume cannot be ignored."

5   (see P3L29-31)

*Line 4.30 Standard notation for ISW plume models has basal freezing as m' and frazil ice growth as f'. Is there a reason that here basal freezing is f' and frazil ice growth is omega'?*

10   Because the basal freezing occurs throughout McMurdo Sound, we think "f'" is more appropriate than "m'" in this study, consistent with "freezing". As Referee #1 suggested, we used "w'" instead of "ω'" to represent frazil ice growth/melting.

*Line 7.11 Describe the type of observations and model results. Something like "... investigated with satellite altimetry and two-dimensional model results." (with the actual adjectives based on the respective, cited papers)*

15   Revised as suggested. However, not all cited papers correspond to two-dimensional models, so we used "numerical models".

[revised manuscript text omitted]

---

## Author Response (AR2)

**2nd round of response to referees:**

**Responses of sub-ice platelet layer thickening rate and frazil ice concentration to variations in Ice Shelf Water supercooling in McMurdo Sound, Antarctica**

Chen Cheng, Adrian Jenkins, Paul R. Holland, Zhaomin Wang, Chengyan Liu, and Ruibin Xia

*This response comprises Reply to Editor, Reply to Referee #1, Reply to Referee #2, and Reply to Referee #3. Each of our replies to the referees is structured as a sequence of comments from the referee, our response, and changes made to the manuscript.*

Note: Italic denotes the referees' comments, and the following is our response. "P*L#" denotes line # in page * of the 'tracked changes' version of the manuscript in which blue characters exactly represent the revised part. Double quote represents the excerpt of the revised manuscript. Almost all the references mentioned in this report can be found in the **References** section of the manuscript; otherwise, we have given the full citation.

**Reply to Editor:**

*Comments to the Author:*
*Dear authors,*

5 *thank you for the revisions of your manuscript which have now undergone another round of thorough reviews. I am glad to say that I agree that I find the manuscript publishable once the new, constructive reviewer's comments have been taken into account and implemented. I am grateful for having received such constructive comments and hope that you will be able to address and implement them almost 1:1. I am looking forward to receive your replies and revisions. Thank you and best regards*

10 *Christian Haas*

Dear Editor Haas,

Thank you for your great support and encouragement for our manuscript. As you suggested, we have taken the new,

15 constructive referees' comments into account and implemented them to our best ability. We hope to address all the specific

issues below. In addition, we clarify that we have carefully checked for typos, missing co-authors and their affiliations,

terminology, updates of data in tables, and updates of variables in equations.

Chen Cheng and other co-authors

**Reply to Referee #1:**

*The authors have revised their manuscript and made significant improvements (including to the quality of the explanations and calculations of the suspension index). The topic of the paper is interesting and worthwhile. However, there remain some substantial issues with the paper, as summarized below. Therefore, I don't think the manuscript presently merits publication. With further changes, it could potentially be published in some journal in future.*

We thank Referee #1 for his/her 2nd round of comments on our manuscript.

*Major issues:*

*Novelty. The authors now do a better job explaining the novelty of their approach and the contribution of previous studies. However, it remains the case that this paper represents fairly incremental progress on their previous modelling paper. The particular geographical application is interesting but not totally compelling (the other reviewers raise important discrepancies between model and observations).*

It was not our intention to develop the earlier model, but rather to apply it to a region where the observations, although still sparse, are the best available to validate the model output. Furthermore, we used the results to investigate the relationship between supercooling and SIPL growth. Therein lies the novelty of our paper. This is stated in the introduction:

"

McMurdo Sound therefore seems an ideal setting in which to apply and evaluate the new vertically-modified ISW plume model proposed by Cheng et al. (2017), which includes time dependence and two horizontal dimensions. The main objective is to explore possibility of finding the quantitative relationship between SIPL thickening rate and ISW supercooling.

"

and we have retained that wording (P3L1-3).

*In terms of interpretation, the paper still does not make it clear why the suspension index should be relatively large (greater than one, say). I can only guess that the various velocities that go into u\* are very small. I could only see one of these velocities reported in Table 1. It is not clear how widely applicable this situation is. For example, if the ambient currents were (say) 0.1 m/s (10 times larger than the value in Table 1), Z\* could be a factor of 10 smaller and the novel process modelled would not be that important. The wider significance of the work is therefore unclear.*

Very small $Z_*$ is just the limiting case of our modification (Fig. 2), which is unlikely to occur in regions of active marine ice/SIPL formation. In our parameterization of frazil ice precipitation rate $p'$ (Eq. 3),

$$p' = w_i C_i \left(1 - \frac{U^2}{U_c^2}\right) \times He\left(1 - \frac{U^2}{U_c^2}\right),$$

the ratio of $\frac{U}{U_c}$ can be regarded as the reciprocal of $Z_*$, that is, $\frac{u_*}{w_i}$ ($\kappa$ is dropped). Therefore, if $Z_*$ is small, $\frac{U}{U_c}$ is large, which limits the precipitation. Therefore, in regions of high precipitation we would expect to find a non-uniform vertical distribution of frazil. We have added the following comment to the manuscript to clarify this point (P4L25-27):

5    "

While low values of $Z_*$ are attainable with strong currents, those conditions also reduce the tendency for frazil to precipitate and contribute to SIPL formation [see Eq. (3) below]. Therefore, we expect a non-uniform vertical distribution of frazil wherever there is active formation of SIPL.

"

*Correctness. A major issue relates to the choice of model parameters and the correctness of the numerical solution of the governing equations. In figure R, the authors present the results of calculations with a better-resolved discretization of the crystal size distribution. These differ markedly from the standard discretization presented in the paper, indicating that the standard results are inaccurate. A reader would not be aware of this important issue from the discussion on P6:L9-13. In*

15    *their response, the authors argue that the poorly resolved discretization is better because it 'agrees' with observation. This argument is fallacious. Rather, the evidence suggests that there is some large flaw in another part of the model (most likely some choice of model parameters or, worse, some structural issue).*

We apologize for a lack of clarity in our earlier response. We agree that the better agreement with observation using the less

20    well resolved crystal size distribution must result from our choice of other parameter values. Those choices are consistent with earlier work (as we now clarify in response to reviewer 2), although most are poorly constrained. The point we were trying to make is that increasing the number of crystal size classes does not alter the results qualitatively (although it does increase the sensitivity to Z*). Therefore, if we were to rerun the models with a better-resolved crystal size distribution and parameters retuned to improve the fit between model results and observations, our conclusions would be the same. We prefer

25    to stick with the simpler model and parameters choices closer to those used by others. To clarify this point we have added the following comment (P6L14-15):

"

Sensitivity experiments with more crystal size classes yielded qualitatively similar results.

"

*The authors have not addressed the issue regarding the conversion of precipitation rate to ice thickening rate, where there are large uncertainties. While outside the scope of their model, this process greatly influences the comparison of the model with observations. I also think that the neglect of latent heat release during volume doubling violates energy conservation.*

Post-depositional processes are beyond the scope of our study, but we have emphasised the importance of considering such processes by adding a citation to the work of Buffo et al (2019) (P7L5-6):

"

Coupling our VM model with a model focusing on the processes associated with platelet ice accretion within the sea ice (Buffo et al., 2018) would be necessary to improve on that rough estimate, but is beyond the scope of the present study.

"

; the associated latent heat release within SIPL should be considered within sea ice models such as that of Buffo et al. (2018). Furthermore, the majority of latent heat released within the sea ice will be lost to the atmosphere above rather than the ocean below.

**Reply to Referee #2:**

*Summary*

*The revisions and corrections made to this manuscript have significantly improved both its interest to the community, and*
5 *clarity for the reader. However, some of my original comments have not been adequately considered, and in my opinion,*
*would need to be properly addressed (both in response to reviewers and by modifying the article text) before the manuscript*
*is ready for publication.*

We thank Referee #2 for his/her 2nd round of constructive comments, which have helped us to improve the clarity of our
10 manuscript.

*Specific points:*
*1. All references to 'temperature' should be changed to 'potential temperature' (assuming this is indeed the quantity used).*
*For example, the text in P2-L15 was modified in response to Reviewer #1's comments, to support the author's assertion that*
15 *temperature can be treated as vertically-uniform within the ISW plume. Unless the qualifier 'potential' is used, this is*
*incorrect, and not supported by observations.*

Revised as suggested

20 *2. Distinction between applicability to crystal accretions beneath sea ice and ice shelves remains unclear. In their response*
*to my review, the authors note that "Beneath the platelet ice layers the supercooling is produced by the pressure drop*
*experience by ISW at it emerges from beneath an ice shelf and rise towards the sea surface, while regions of marine ice*
*accretion beneath ice shelves have typically very low basal slopes." I completely agree with this, and am therefore puzzled*
*as to why the authors seem reluctant to identify this difference to the reader. This is exactly the mechanistic driver that*
25 *would allow the sub-ice boundary layers to behave differently.*
*I am not suggesting that the work (tested for platelet layers beneath sea ice, and potentially applied to marine ice layers*
*beneath ice shelves) is invalid. However, I do think that a statement clarifying the differences between the regimes is*
*necessary for the reader (and may protect the authors of the present study in the event that their model is naively applied to*
*a regime for which it has not been validated). Specifically, the potential sources of divergence are:*
30 *a. The difference in basal slope is likely to result in the ISW plume existing much closer to the in-situ freezing temperature*
*than is observed in McMurdo Sound, with the result that in-situ supercooling is likely to be much smaller (and potentially*
*similar to the resolution of present-day instruments. i.e. unobservable except by implication) beneath ice shelves;*

*b. The difference in basal slope also has implications for generating buoyancy-induced momentum, which is the implicit source of the background current in Hughes et al. (2014), and therefore would need to be excluded (or vastly reduced) for application to an ice shelf cavity.*

*c. The authors allude to another difference with their statement "The one indirect effect on the ISW plume might be an*
5 *increased drag coefficient beneath the platelet layer, where more rapid freezing of the deposited crystals may create more irregularity in the form of the ice-ocean interface." This is true, and will affect the sedimentation process.*

*d. In addition to the above, the length of time over which crystal accretion may occur is vastly different (i.e. about 1-3 years in McMurdo Sound vs tens-hundreds of years beneath ice shelves). Combined with the likely difference in degree of supercooling, this could potentially lead to very different internal structures of the crystal layers (e.g. marine ice layers more*
10 *likely to collapse under accreted buoyancy), and hence present different physical boundaries to ocean flow, and an entirely different source of effective hydrodynamic drag (as suggested by Robinson et al., 2017, which the authors cite).*

*e. Finally, the observed supercooled plume in McMurdo Sound, having only recently experienced the step-change in pressure, is still adjusting to the change through active ice formation (onto both suspended frazil and to accreted platelet ice), and will therefore come to a point of equilibrium at some point beyond where the observations to date have been made.*
15 *This represents a significantly different thermodynamic regime to the general situation of an ISW plume beneath an ice shelf.*

*This specifically relates to P11L8-10 of the revised manuscript, since in the general sub-ice shelf regime, the supercooled layer will almost certainly not approach the thickness observed in McMurdo Sound. This may have implications for the regime for which 'the efficiency of converting ISW supercooling into frazil concentration ... is determined by the suspension*
20 *index', since this is true only when 'the thickness of a supercooled layer of ISW is large enough' (P11L8-10) (i.e. greater than 65 m for the McMurdo Sound parameters).*

*I suggest that the addition of a well-crafted paragraph of text outlining the potential differences between the regimes would be sufficient to both demonstrate that the authors understand the implications of these differences (and I am convinced they*
25 *do), and highlight to the reader where caution (and/or improved understanding) is required in applying this model to the sub-ice shelf regime.*

Thank you for your very thorough discussion of this specific issue. We added more details as suggested (P12L10-25):
"
30 Results may differ from those discussed above, because of the subtly different environments beneath sea ice and ice shelves. Beneath a SIPL, supercooling is produced by the pressure drop experienced by ISW as it emerges from beneath an ice shelf and rises towards the sea surface, while supercooling that drives marine ice accretion beneath ice shelves is produced as the ISW ascends a very gentle basal slope. The in-situ supercooling level beneath ice shelves is therefore likely to be much smaller than that observed in McMurdo Sound, while the differing slopes also yield differing buoyancy forcing on the flow.

Furthermore, after experiencing the step-change in pressure as it ascends the ice front, the supercooled plume in McMurdo Sound is in the process of adjustment, through the formation of suspended frazil and direct freezing onto the accreted SIPL, towards an equilibrium that is presumably attained beyond the region of observations. At the base of an ice shelf, typically several hundred meters thick, the vertical temperature gradient is comparatively small, so the deposited crystals form a

5 slushy layer (Engelhardt and Determann, 1987) that slowly consolidates, possibly as much through compaction as freezing. The ice-ocean interface and the associated drag coefficient are therefore likely to be very different to those observed in McMurdo Sound, where SIPL appears to comprise a more open matrix of ice and water that consolidates by freezing as heat is lost to the atmosphere. In addition, the vastly different time scales over which crystal accretion occurs (about 1-3 years in McMurdo Sound vs tens-hundreds of years beneath ice shelves) could lead to further differences in the internal structure of

10 the crystal layers and hence in the physical boundaries they present to the ISW plume.
,,

3. *Justification for values of parameters used, and acknowledgement of available observations. In their response, the authors point to the lack of 'observational guidance' as justification for the extensive tuning of specific parameters. I agree, there*

15 *are very little data available to constrain the models. However, the values they have chosen do find support in the literature, and it would strengthen the paper to acknowledge these. In particular (P7L9-10):*

*a. ISW outflow properties: the chosen values for temperature and salinity coincide with those reported by Hughes et al., 2014;*

*b. Platelet layer basal drag coefficient: the value chosen fits appropriately within the range identified by Robinson et al.,*

20 *2017;*

*c. Frazil ice crystal size distribution: unknown, but presumably these are chosen from somewhere – perhaps previous modelling studies? Similarly for the Shields criterion?*

*d. Ambient current speed: The chosen value is consistent with the lowest speeds reported by Robinson et al., 2014 (although lower than their reported residual flow).*

25 *e. In addition, the observations in both Hughes et al. (2014) and Robinson et al. (2014) papers show the homogeneous ISW layer (observed in the centre of the modelled plume flow) as being 150 - 200 m thick, and the supercooled portion extending to 60/70 m. I would have thought these would be useful reference points for this manuscript.*

Thank you for these useful comments. We added more details as suggested (P7L15-23):

30 "

Despite the limited observational constraints on many of these parameters we do find support in the literature for our adopted values: ISW outflow properties are consistent with those reported by HU14, and the corresponding thickness of supercooled layer is within the observed range (60-70 m) given in both HU14 and Robinson et al. (2014); the basal drag coefficient fits appropriately within the range identified by Robinson et al. (2017), while the ambient current speed is consistent with the

lowest speeds reported in that study; we used 5 crystal size classes, as did Galton-Fenzi et al. (2012), although our sizes are slightly larger; we used a larger Shields criterion than the middle (0.05) of the observed range, although there is considerable scatter amongst the individual results reported from sedimentary experiments. Table 1 summarises all the values adopted for the key parameters.

5    "

*4. My concern about the apparent resolution in figures 5 & 6 still stands: the separation of the contour lines, especially around the core of the plume, implies greater resolution than the model contains. A potential solution may be to plot only every second contour line.*

As suggested, we halved the number of contour lines.

**Reply to Referee #3**

*The paper is logically and clearly written. It describes the application of a two-dimensional ISW plume model to a set of observations of an ISW under sea ice. The results are an improvement over the model of Hughes et al (2014), particularly in terms of their two-dimensional nature that illustrates the Coriolis effect along the centre line of the plume. In addition the authors show (as naively might be expected) that the vertical distribution of supercooling and frazil concentration determine the growth of the sub-ice platelet layer.*

We thank Referee #3 for his/her constructive comments that have helped us to improve our manuscript.

*Comment 1: Some details of the process of formation of sea ice need to be outlined and are ignored in the paper. There are two consequences*

*(i) The term "platelet layer" is confusing. The term "sub-ice platelet layer" was coined by Gow et al (1998) and later authors for the high porosity layer that principally forms by accumulation of ice from the water column. I think the authors use "platelet layer" to mean this friable layer beneath the more consolidated "incorporated platelet ice". Incorporated platelet ice can be identified easily in an ice core by its crystal structure, but its other physical properties (salinity, porosity, permeability) are very similar to the usual columnar sea ice. It is formed by the freezing of the sub-ice platelet layer due to heat loss to the atmosphere. The term "platelet layer" could be understood to mean the sum of the incorporated platelet ice and the sub-ice platelet layer. In my view "sub-ice platelet layer" or "SIPL" (as in the original version) should be used to avoid confusion.*
*Gow, A. J., S. F. Ackley, J. W. Govoni, and W. F. Weeks (1998), Physical and structural properties of land-fast sea ice in McMurdo Sound, Antarctica, in Antarctic Sea Ice: Physical Processes, Interactions and Variability, Antarct. Res. Ser., vol. 74, edited by M. O. Jeffries, pp. 355–374, AGU, Washington, D. C., doi:10.1029/AR074p0355.*

We agree with this point. First, we added a citation to Gow et al. (1998) where the phrase "sub-ice platelet layer" first occurs (P2L3). Second, the process by which the upper parts of the SIPL become incorporated into the sea ice has been described in the previous version of the manuscript (P2L4-6 in the new revised manuscript). Third, we have reverted to the use of the acronym SIPL instead of the term "platelet layer" in both text and figures.

*(ii) The data used from HU14 is the "sub-ice platelet layer" thickness in November. However the top portion of this layer will have become incorporated into the consolidated sea ice above. Thus the value used will be an underestimate of the true value had the freezing due to heat loss to the atmosphere not taken place. Better agreement might be obtained between model and observation if the thickness of the incorporated platelet ice was included. There are some values of the*

*incorporated platelet ice thickness in HU14 (Fig 8b) but it would be better derived from Figs 2 & 3 and the Supplementary material of Langhorne et al (2015).*

We admit the incorporation of SIPL into the sea ice adds additional uncertainty into the comparison of our simulated SIPL
5   thickness with observations. However, the complex processes by which SIPL is incorporated into the sea ice cover are beyond the scope of our study. The assumption we make in converting frazil precipitation into SIPL thickness and our direct comparison of that result with measured SIPL thickness follow exactly the procedures of HU14. This seems the most consistent approach, although, as stated in the new revised manuscript (see P6L31-P7L6), we believe that this issue would be potentially improved by coupling with specific sea ice models focusing on microscale processes within the bottom layer of
10   the ice, such as Buffo et al. (2018).

*Comment 2: The authors have conducted a significant sensitivity study. I would like to know what can be learned from this sensitivity study about the physics controlling the process of deposition of ice crystals; and how could this inform the design of observational campaigns. For example Fig 6 shows much steeper gradients in sub-ice platelet layer thickness than*
15   *demonstrated in the observations. Why? Could there be post-depositional processes taking place that are not accounted for in the model; or a mismatch in the deifnition of between modeled and observed "platelet layer" (see next comment) or undersampling of the observations.*

*The authors are rather glib in simply saying that small scale features were not resolved by relatively coarse scale spatial*
20   *distribution of the sea ice thickness measurements (incidently these were not ice core samples). Experimental evidence of steep gradients in the sub-ice platelet layer thickness have been published (Hunkeler et al, 2015) but these were associated with a sea ice breakout and an abrupt change in sea ice plus snow thickness. A significant change in sea ice thickness was not observed in the McMurdo Sound observations.*

25   *Hunkeler, P. A., M. Hoppmann, S. Hendricks, T. Kalscheuer, R. Gerdes (2016), A glimpse beneath Antarctic sea ice: Platelet layer volume from multifrequency electro- magnetic induction sounding, Geophys. Res. Lett., 43, 222–231, doi:10.1002/2015GL065074.*

*Further if the position of the 3 km wide plume outflow (an unknown in the model) was moved east or west by a small amount*
30   *then there would be a big discrepancy between model and experiment.*

The main purpose of conducting the extensive sensitivity runs was to reveal a comprehensive area-averaged relationship between SIPL thickening rate (i.e., the process of deposition of ice crystals) and ISW supercooling in McMurdo Sound, and that was shown to be predominantly controlled by the suspension index. Therefore, we highlight the need to improve the

detection technology for suspended ice crystals within the sub-ice oceanic boundary layer. Referring to the steep gradient of SIPL thickness, HU14 also obtained a comparable gradient, that is, a decrease from 17 to 9 m (15 to 6 m in this study) within 5 km of the outflow (HU14, Fig. 10e). We have now expanded on the possible reasons why simulated gradients might be greater than observed ones (P8L25-P9L4):

5   "

The simulated SIPL thickness near the ISW outflow exhibits steeper gradients than are observed (Fig. 6b and c), which probably result from the spatial non-uniformity of ISW plume near the outflow (Fig. 5a and b). That non-uniformity in flow leads to localized non-uniformities in thermodynamics (Fig. 5c and d), frazil concentration (Fig. 5e and f), and thus SIPL thickness (Fig. 6b and c). Moreover, because the sea ice base is horizontal, there are no changes in the freezing point

10   associated with pressure change, so supercooling is always highest at the ISW outflow (Fig. 5c and d). That results in the greatest frazil concentration (Fig. 5e and f) and SIPL thickness (Fig. 6b and c) near the location of the outflow, and because the outflow is steady in time spatial gradients in SIPL close to the outflow are enhanced. In reality, temporal changes in the ISW outflow position, width, supercooled layer thickness and duration could lead to a broader region of elevated frazil precipitation and a less peaked distribution of SIPL thickness. In addition, such small-scale features in the SIPL thickness

15   distribution, if present, would not be resolved by the relatively coarse spatial distribution of drill-hole measurements (dots in Fig. 6). Nevertheless, the largest SIPL thickness undoubtedly occurs adjacent to the ISW outflow in McMurdo Sound, and the SIPL thickness calculated by the VM model at drill sites agrees well with the measurements (Fig. 6a), being graded "excellent" in contrast with the "poor" performance of the VU model (Table 2).
"

20   We also revised the text (P6L5-6) as "The initial thickness of the ISW outflow (indicated by blue arrow in Fig. 1) from underneath McMurdo Ice Shelf …" to clarify the position of the ISW outflow, and used the phrase "drill-hole measurements" instead of ice core samples after scrutinizing HU14 and Price et al. (2014) (Price, D., Rack, W., Langhorne, P. J., Haas, C., Leonard, G., and Barnsdale, K.: The sub-ice platelet layer and its influence on freeboard to thickness conversion of Antarctic sea ice, The Cryosphere, 8, 1031-1039, https://doi.org/10.5194/tc-8-1031-2014, 2014.). The latter utilized the same data sets

25   as in HU14 and the present study.

*Comment 3: Fig 1 is incorrect. A pre-2011 ice shelf front has been used and this means that the purple box has been placed too far to the north. The samples stations (red dots) are in the wrong locations.*

30   Thank you for this correction. We have relocated the purple box and sample stations (red dots) completely following Figs. 6 and 9 in HU14.

*Technical Corrections*

*p. 2, line 15-24: I was surprised that the instabilities derived by Jordan et al (2014) were not discussed in relation to the vertical distribution of frazil. Are they not relevant?*

Jordan et al. (2014) (Jordan, J. R., Kimura, S., Holland, P. R., Jenkins, A., and Piggott, M. D.: On the conditional frazil ice instability in seawater, J. Phys. Oceanogr., 45, 1121-1138, http://doi.org/10.1175/JPO-D-14-0159.1, 2015) used a non-hydrostatic ocean model to study the conditional frazil ice instability during the rise of frazil ice from a depth of several hundred metres to the sea surface, while this study is focused on depth-integrated sub-ice ISW plume modelling that considers an equilibrium vertical distribution of frazil concentration within the plume. Therefore, although both studies are associated with frazil ice processes, they are not closely-related.

*p. 3, Line 2-3: "The main objective is to quantify for the first time the response of the platelet layer thickening rate to variations in ISW supercooling." I don't think this can be claimed as a first as HU14 considered supecooling and sub-ice platelet layer thickness, as did Buffo et al (2018). Please delete "for the first time"*

*p. 8, line 26-30: "we know of no studies to date of the response of marine ice (or platelet layer) thickening rate beneath ice shelves (or sea ice) to variations in supercooling," Again I would argue that HU14 and Buffo et al (2018) considered this relationship.*

We admit that both HU14 and Buffo et al. (2018) focused on the issue of SIPL growth from supercooled water, but neither explored the quantitative relationship between ISW supercooling and SIPL thickening rate. We have edited the sentences to clarify the novel aspect of our study:

"

The main objective is to explore possibility of finding the quantitative relationship between SIPL thickening rate and ISW supercooling.

" (P3L2-3).

"

In contrast, we know of no studies to date that provide a quantitative relationship between marine ice (or SIPL) thickening rate beneath ice shelves (or sea ice) and ISW supercooling. Such a relationship is of potential significance for evaluating the mass balance of deep-draughting ice shelves in cold water environments and adjacent sea ice subject to climatic variability.

" (P9L15-18).

*p. 8, line 13-18: The authors are rather glib in simply...there would be a big discrepancy between model and experiment.*

See the above response to Comment 2.

*On the other hand I think the authors could expand on the relationship and provide insights and explanations.*

*p. 10, line 21-26: This is exactly the sort of explanation that I find a useful outcome of the detailed modeling.*

The relationship is rather complex, and we have attempted to offer insight through our choice of Figures. Fig. 7a depicts the relationship between ISW supercooling and thickening rate as a function of supercooling of the outflow, indicating two critical supercooling levels dividing the relationship into three regimes. Fig. 7b demonstrates that this complexity stems from the suspension index. Fig. 8 further illustrates the control of suspension index on the efficiency with which the supercooling is utilized. Fig. 10 more explicitly presents the control of suspension index on the thickening rate and frazil concentration. Subsections 4.2 and 4.3 discuss these figures in detail.

[revised manuscript text omitted]